



# Insights into the single particle composition, size, mixing state and aspect ratio of freshly emitted mineral dust from field measurements in the Moroccan Sahara using electron microscopy

Agnesh Panta[1], Konrad Kandler[1], Andres Alastuey[2], Cristina González-Flórez[3], Adolfo González-Romero[2,3], Martina Klose[4], Xavier Querol[2], Cristina Reche[2], Jesús Yus-Díez[2,5], and Carlos Pérez García-Pando[3,6]

[1]Atmospheric Aerosol Group, Institute of Applied Geosciences, Technical University of Darmstadt, Darmstadt, Germany
[2]Institute of Environmental Assessment and Water Research (IDAEA-CSIC), Barcelona, 08034, Spain
[3]Barcelona Supercomputing Center, Barcelona, Spain
[4]Department Troposphere Research, Institute of Meteorology and Climate Research (IMK-TRO), Karlsruhe Institute of Technology (KIT), Karlsruhe, Germany
[5]Grup de Meteorologia, Departament de Física Aplicada, Universitat de Barcelona, C/Martí i Franquès, 1, 08028, Barcelona, Spain
[6]ICREA, Catalan Institution for Research and Advanced Studies, Barcelona, Spain

**Correspondence:** Agnesh Panta (agnesh.panta@tu-darmstadt.de)

**Abstract.** The chemical and morphological properties of mineral dust aerosols emitted by wind erosion from arid and semi-arid regions influence climate, ocean and land ecosystems, air quality, and multiple socio-economic sectors. However, there is an incomplete understanding of the emitted dust particle size distribution (PSD) in terms of its constituent minerals that typically result from the fragmentation of soil aggregates during wind erosion. The emitted dust PSD affects the duration of particle transport and thus each mineral's global distribution, along with its specific effect upon climate. This lack of understanding is largely due to the scarcity of relevant in situ measurements in dust sources. To advance our understanding of the physicochemical properties of the emitted dust PSD, we present insights into the elemental composition and morphology of individual dust particles collected during the FRontiers in dust minerAloGical coMposition and its Effects upoN climaTe (FRAGMENT) field campaign in the Moroccan Sahara in September 2019. We analyzed more than 300,000 freshly emitted individual particles by performing offline analysis in the laboratory using Scanning Electron Microscopy (SEM) coupled with Energy-Dispersive X-ray Spectrometry (EDX). Eight major particle-type classes were identified where clay minerals make up the majority of the analyzed particles by number, with carbonates and quartz contributing to a lesser extent. We provide an exhaustive analysis of the size distribution and potential mixing state of different particle types, focusing largely on iron-rich (Fe-oxi/hydroxides) and feldspar particles, which are key to the effects of dust upon radiation and clouds. Nearly pure or externally mixed Fe-oxi/hydroxides are present only in diameters smaller than 2 $\mu$m and mainly below 1 $\mu$m. Fe-oxi/hydroxides tend to be increasingly internally mixed with other minerals, especially clays, as particle size increases, i.e., the volume fraction of Fe-oxi/hydroxides in aggregates decreases with particle size. Pure (externally-mixed) feldspar represented 3.7 % of all the particles, of which we estimated about a quarter to be K-feldspar. The externally-mixed total feldspar and K-feldspar abundances are relatively invariant with particle size, in contrast to the increasing abundance of feldspar-like (internally-mixed) aggregates





with particle size. We also found that overall the median aspect ratio is rather constant across particle size and mineral groups, although we obtain slightly higher aspect ratios for internally-mixed particles. The detailed information on the composition of freshly emitted individual dust particles along with the quantitative analysis of their mixing state presented here can be used to constrain climate models including mineral species in their representation of the dust cycle.

# 1   Introduction

Mineral dust, as one of the most important natural aerosols in the atmosphere, is a key player in influencing the global climate system (Shao et al., 2011). It is the most abundant aerosol type in terms of dry mass (Choobari et al., 2014; Textor et al., 2006), with an estimated emission flux between 3400 and 8900 Tg yr$^{-1}$ for particles smaller than 20 $\mu$m geometric diameter (Kok et al., 2022). In particular, Northern Africa is responsible for an estimated 50 % of global windblown mineral dust emissions (Engelstaedter et al., 2006; Kok et al., 2021). Dust affects directly the Earth's radiative budget via scattering and absorption

of radiation (Sokolik et al., 2001; Pérez et al., 2006; Strong et al., 2018), and indirectly by influencing cloud development on a microphysical level by acting as cloud condensation nuclei and ice nuclei (Zimmermann et al., 2008; Kumar et al., 2011; Hoose and Möhler, 2012; Froyd et al., 2022), and thus impacting cloud optical properties and radiation. Once uplifted over the source areas, dust can travel thousands of kilometers away from these and can act as a carrier for micronutrients, such as iron and phosphorous, to the ocean (Jickells et al., 2005; Schulz et al., 2012; Rodríguez et al., 2021; Myriokefalitakis et al., 2022)

and the Amazon rainforest (Yu et al., 2015), with implications for biogeochemical cycles and ocean uptake of atmospheric carbon dioxide by promoting phytoplankton growth (Mahowald et al., 2009). Additionally, dust can be deposited on snow or ice thereby reducing snow/ice reflectance (surface darkening) and consequently changing the climate and water cycle by accelerating snow melting (Painter et al., 2010; Sarangi et al., 2020). Dust can also interact with air pollutants by acting as a catalyzer and providing a surface for heterogeneous reactions (Cwiertny et al., 2008; Ndour et al., 2008). Finally, dust might

negatively impact human health (Querol et al., 2019; Pérez García-Pando et al., 2014; Giannadaki et al., 2014), induce a decrease in solar energy yield by dust deposition to solar panels (Piedra and Moosmüller, 2017), and negatively impact aircraft operations due to reduced visibility (Middleton, 2017; Monteiro et al., 2022). Most of these processes rely not only on the total mass of dust but also on its microphysical properties and composition (Mahowald et al., 2014).

    The physicochemical properties of mineral dust aerosols including their particle size distribution (PSD), composition, mix-

ing state and shape determine their impact on climate and atmospheric chemistry (Formenti et al., 2011). The emitted PSD, spanning from a few nanometers to hundreds of micrometers, is critical in constraining the dust atmospheric lifetime, as coarser particles tend to deposit quickly due to gravity, albeit less than previously thought (van der Does et al., 2018; Adebiyi and Kok, 2020; Adebiyi et al., 2022). The chemical and mineralogical composition is another key factor to consider; for example, the absorption properties and ice-nucleating ability of dust depend upon its mineralogy, mainly upon the presence of iron oxides

(Li et al., 2021) and K-feldspar (Atkinson et al., 2013; Kiselev et al., 2017; Welti et al., 2019; Yun et al., 2020), respectively. The mixing or aggregation state of the different dust minerals is also important in determining the behavior of dust in the atmosphere in terms of its reactivity, chemical processing, optical properties, ice nucleation ability and dust deposition (Fitzgerald





et al., 2015; Kandler et al., 2018). For example, dust is more absorbing when iron oxides are internally mixed with other minerals than when they are externally mixed (Sokolik and Toon, 1999). The feldspar fraction in desert soils also varies considerably. In parts of northern Africa (southern Algeria, northern Mauritania and northern Niger) it can reach $\sim$ 12 to 20 % whereas, in western Sahel, the feldspar fraction is less than 2% (Nickovic et al., 2012; Perlwitz et al., 2015a). In addition, the ice nucleation property of mineral dust depends not only on the source region but also on the feldspar size distribution, which among other aspects, dictates its lifetime in the atmosphere. Moreover, the ice-nucleating property of K-feldspar varies significantly despite having a comparable crystal structure and composition, leading to a variety of ice-nucleating abilities (Harrison et al., 2016). K-feldspar also dominates the number of ice nuclei in both the internally mixed and externally mixed cases (Atkinson et al., 2013). Particle shape influences the dust single-scattering properties (Lindqvist et al., 2014; Nousiainen and Kandler, 2015; Saito and Yang, 2021). Accurate quantification of dust shape is important for calculating the dust impact on radiative forcing (Ito et al., 2021) and the terminal velocity of dust particles (Ginoux, 2003; Huang et al., 2020), albeit with high uncertainties for both of these effects (Nousiainen et al., 2011). Mineral dust also contributes significantly to the atmospheric aerosol mass loadings which are retrieved from satellite measurements and ground-based lidar measurements applying algorithms that use particle shape as one of the input parameters (Dubovik et al., 2006; Gliß et al., 2021). The importance of realistic size equivalence and shape of spheroidal Saharan dust particles on optical properties and the radiative effect was studied in Otto et al. (2011) with data gathered during SAMUM-1 in Morocco. Furthermore, scanning electron microscopy (SEM) was used to characterize the mineralogical composition and shape of mineral dust particles collected during the SAMUM campaign over Morocco in 2006 (Lindqvist et al., 2014). They found great variation between the scattering properties of spheres, spheroids, and mineral dust particles characterized by SEM (Lindqvist et al., 2014). Moreover, for non-symmetrical particles, a preferential orientation in the atmosphere is observed where a shape-dependent settling behavior comes into effect (Li and Osada, 2007). There is also a shape-dependent separation of mineral dust into layers of different altitudes during cross-Atlantic transport of Saharan dust which strongly points to a shape-preferential settling of particles (Yang et al., 2013). In addition, dust nonsphericity could enhance the snow albedo reduction by up to 20 % relative to spherical dust (Shi et al., 2022). As the shape of dust particles is highly aspherical (Huang et al., 2020), this can further influence their optical properties (Ito et al., 2021; Otto et al., 2009; Mishchenko et al., 1997; Nousiainen and Kandler, 2015; Klose et al., 2021). Finally, the dust lifetime in the atmosphere is also affected by particle shape and density (which depends on mineralogy), with more spherical and denser particles being deposited faster (Huang et al., 2020; Mallios et al., 2020). Other potentially important aspects that have been largely unexplored are the potential interdependencies among size, composition, mixing state, and shape. For example, while it is well known that composition is size-dependent (Kandler et al., 2009, 2011; Ryder et al., 2018; Liu et al., 2018), very little is known about the potential dependencies of shape and mixing state upon both composition and size.

Models that include spatiotemporal variations in mineralogical composition are relatively new (Perlwitz et al., 2015a, b; Scanza et al., 2015) and currently use rather crude soil mineralogy maps (Claquin et al., 1999; Nickovic et al., 2012; Journet et al., 2014) as a lower boundary condition. Soil mineralogy maps are based on massive extrapolation from a limited amount of soil mineralogical analyses, ancillary information on soil texture and color, and several additional assumptions. This limited knowledge together with our incomplete understanding and scarcity of measurements on the emitted dust physicochemical





properties and their relationship with the PSD and composition of the parent soil precludes accurate model assessment of the effects of dust upon climate (Perlwitz et al., 2015a; Pérez García-Pando et al., 2016; Li et al., 2021). One reason for such measurement scarcity is that source areas, particularly the host prolific ones, are typically located in remote areas and subject to harsh conditions. Also, frequent dust storms can lead to large dust concentrations posing a great challenge to sample for single-particle analysis as the filter can be quickly overloaded.

In this contribution, we investigate the size, morphology, elemental (and mineralogical) composition, and mixing state of freshly emitted individual dust particles based on samples collected during a major wind erosion and dust emission field campaign in the Moroccan Sahara within the framework of the FRontiers in dust minerAloGical coMposition and its Effects upoN climate (FRAGMENT) project. Three different sampling instruments were used to collect freshly emitted mineral dust particles and the chemical and physical properties of individual dust particles were examined in detail using SEM coupled with an energy-dispersive X-ray analysis (EDX). Our detailed chemical and physical speciation represent a first step towards advancing our knowledge on the emitted PSD of individual dust minerals, that will ultimately help understanding the relationship between the size-resolved composition of the emitted dust and that of the parent soil. Such knowledge is needed to better constrain climate models that are starting to consider mineralogical variations in their representation of the dust cycle (Perlwitz et al., 2015a; Scanza et al., 2015; Li et al., 2021), and it is timely given the prospect of global soil mineralogy retrievals using high-quality spaceborne hyperspectral measurements (Green et al., 2020).

## 2 Materials and methods

### 2.1 Measurement site

In situ aerosol sampling was conducted in a small study area in SE Morocco during FRAGMENT wind erosion and dust emission field campaign between 03 September and 01 October 2019. A suite of meteorological and aerosol instruments was deployed as depicted in Fig. 1 to measure key meteorological and aerosol quantities. Below we describe only the instruments and measurements used in this contribution. Measurements performed during the campaign with other instruments displayed in Fig. 1b are discussed in companion papers (González-Flórez et al., in prep.; Yus-Díez et al., in prep.). The study area, locally known as L'Bour (29°49'30" N, 5°52'25" W ≈ 500 m a.s.l.), is a small ephemeral lake located in the Lower Drâa Valley of Morocco and lies at the edge of the Saharan Desert approximately 15 km to the west of the small village M'Hamid El Ghizlane. The region is characterized by high aerosol optical depth (Ginoux et al., 2012). The location was chosen primarily based on its dust emission potential and logistical feasibility. L'Bour is approximately flat and devoid of vegetation or other obstacles within a radius of ∼1 km and is surrounded by small sand dune fields. The surface consists of a smooth hard crust (paved sediment) mostly resulting from drying and aeolian erosion of paleo-sediments that is analyzed in detail in a companion paper (González-Romero et al., in prep.). Dust is emitted frequently from here in favourable emission conditions (González-Flórez et al., in prep.). PSDs of the paved sediment were analyzed using dry dispersion (minimally dispersed) and wet dispersion (fully dispersed) techniques and displayed two prominent modes at $\sim 100$ $\mu$m and $\sim 10$ $\mu$m (González-Romero et al., in prep.). According to the fully dispersed PSD, the texture of the surface paved sediment is loam (Valentin and Bresson, 1992).





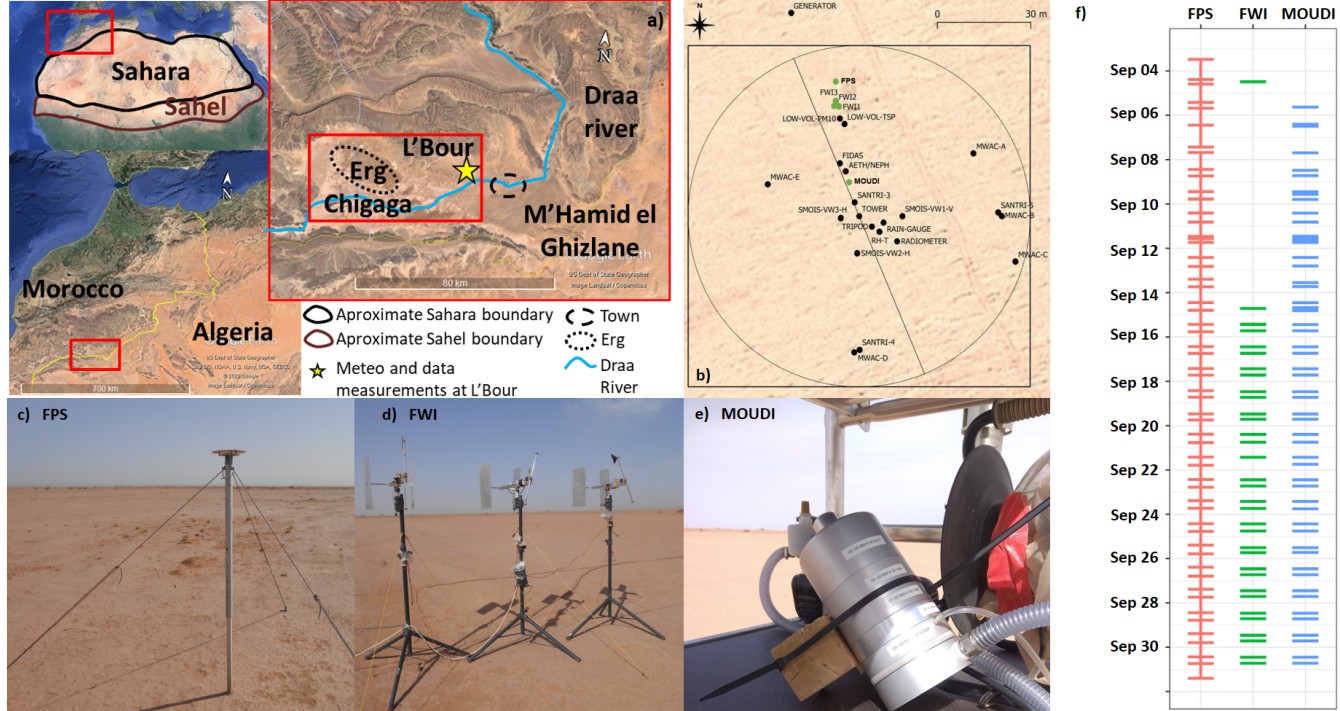

**Figure 1.** (a) Northwestern Africa showing area of operation of FRAGMENT campaign. The main location L'Bour is shown by a star sign which is located close to the Drâa river basin. (b) Schematic representation of different instruments used in the campaign with green dots highlighting those instruments used for single particle analysis. (c) Flat-plate deposition sampler (FPS). (d) Free-wing impactor (FWI). (e) Cascade impactor (MOUDI). (f) Sampling schedule of particle collection using different techniques. The horizontal bar represents start and end of sampling time. Note that due to very short sampling time for FWI and MOUDI, the two bars (start and end) appear as one. Image (a) was created using © Google Earth (US Dept of State Geographer, Data SIO, NOAA, US Navy, NGA, GEBCO, ©2020 Google, Image Landsat/ Copernicus).

## 2.2 Sampling techniques

Particles were collected near ground level (∼ 1.5 m) using three different sampling techniques, namely: flat-plate sampler (FPS), free-wing impactor (FWI), and a micro-orifice uniform deposit impactor (MOUDI, MSP Corp. Model 110) all of which are briefly described in the section below. A detailed description and methodology of sampling techniques have been provided

in previous publications (Kandler et al., 2018; Waza et al., 2019). All aerosol samples were collected on pure carbon adhesive substrates (Spectro Tabs, Plano GmbH, Wetzlar, Germany) mounted on standard SEM aluminum stubs. Pure carbon substrate was chosen because of its excellent smooth surface resulting in a clear image-analytical separation of particles and background (Ebert et al., 1997). All adhesive samples were stored in standard SEM storage boxes (Ted Pella Inc., Redding, CA, USA) in dry conditions at room temperature. Substrates in the MOUDI and FWI were collected two times a day with a typical sampling





duration of a few minutes to avoid overloading the substrate for individual particle analysis. For the flat-plate sampler, the
average exposure time was half a day.

### 2.2.1   Free-wing Impactor (FWI)

A free-rotating wing impactor (Kandler et al., 2009, 2018; Waza et al., 2019) was used to collect particles larger than ~3
μm (projected diameter). The FWI has a sticky carbon substrate as impaction surface as is attached to a rotating arm that
moves through the air; particles deposit on the moving plate due to their inertia. The rotating arm is moved at a constant speed
by a stepper motor, which is fixed on a wind vane, aligning the FWI to the wind direction. The substrate itself is oriented
perpendicular to the air stream vector (resulting from wind and rotation speeds) by a small wind vane attached to the rotating
arm. The particle size cutoff is defined by the impaction parameter, i.e., by rotation speed, wind speed, and sample substrate
geometry.

### 2.2.2   Flat plate sampler (FPS)

The flat-plate sampler used in this work is based on the original flat-plate sampler used in Ott and Peters (2008). It consists of
two round brass plates (top-plate diameter of 203 mm, bottom plate 127 mm, thickness 1 mm each) mounted with a distance
of 16 mm separating them. The plates protect the substrate from precipitation and reduce the effects of wind speed by reducing
the smallest turbulences to the distance between the parallel plates. In this setup, larger particles ($> 1$ mm) cannot enter the
sampling stub surface at low wind speed (Ott and Peters, 2008). The main triggers for particle deposition on the substrate are
diffusion, gravity settling, and turbulent inertial forces.

### 2.2.3   Cascade impactor (MOUDI)

Sampling was conducted using a 5-stage Micro-Orifice Uniform Deposit Impactor (MOUDI, MSP Corp., MN, USA) with a
100 L min$^{-1}$ flow rate. The impactor has stages available with 50% cut-point aerodynamic diameter ($D_a$) of 10, 2.5, 1.0, and
0.25 μm. Particles were collected on 25-mm sticky carbon substrates attached to the 75-mm impaction plate using a double-
sided adhesive. The sampling time was chosen to provide optimum particle loading for single particle analysis which was
usually limited to a few minutes. The short sampling time resulted in fewer particle collections in the first two stages and
therefore only the third and fourth stages are chosen for detailed analysis.

Particle bounce can be an issue with an inertial impactor such as the MOUDI and occurs when particles are impacted on the
collection substrate but are not retained. Studies have shown that particle bounce reduces at the relative humidity of $> 70$ %
depending on the chemical composition of particles (Bateman et al., 2014; Fang et al., 1991; Vasiliou et al., 1999). During the
measurement campaign, the relative humidity was usually below 60 % with a maximum of 71 %. Therefore, particle bounce
can occur which could lead to biases in the particles collected. Additionally, the shattering and asphericity of dust particles can
further induce bias in collected particles.





## 2.3 Scanning electron microscopy (SEM)

Size, morphology, and elemental composition of a large number of individual particles were investigated by SEM as done in previous studies such as (Kandler et al., 2007; Chou et al., 2008; Ryder et al., 2018; Kandler et al., 2020). About 64,000 (FWI), 100,000 (FPS), and 176,000 (MOUDI) individual particles were analyzed with a scanning electron microscope (FEI ESEM Quanta 400 FEG instrument, Eindhoven, The Netherlands) equipped with an X-Max 150 Energy-Dispersive X-ray spectroscopy (EDX) silicon drift X-ray detector (Oxford, Oxfordshire, UK). The samples were analyzed in high vacuum conditions without any pretreatment and were automatically detected using computer-controlled SEM with EDX (Oxford software AZtec 4.4). Backscattered electron (BSE) image was used for particle identification as dust particles are composed of elements with atomic numbers greater than carbon and therefore visible as detectable bright spots in the BSE image. An acceleration voltage of 12.5 kV, beam current of 18 nA, "spot size 5.0" (beam diameter of $\sim$ 3 nm), and a working distance of approximately 10 mm were used to produce the optimum number of input counts in the EDX detector. The scanning resolution was tuned to particle size. For the MOUDI and FPS, 160 nm per pixel was used to identify particles up to 0.5 $\mu$m projected area diameter depending on the sample and for the FWI, 360 nm per pixel was used to identify the largest particles (mainly particles larger than 2.5 $\mu$m in projected area diameter). Chemistry information is derived by EDX. The internal ZAF correction (Z – atomic number, A – absorption, F – fluorescence, accounting for material-dependent efficiencies) of the detector/software system–based on inter-peak background radiation absorption measurements for correction–was used for obtaining quantitative results. The SEM-EDX analysis normalizes the elemental composition to 100 % among the selected elements (C, N, O, Na, Mg, Al, Si, P, S, Cl, K, Ca, V, Cr, Mn, Fe, Zn, and Pb). The detection limits of each element are determined based on $2\sigma$ of the peak intensity and calculated with the AZtec software. A final sorting step is done to remove particles with low X-ray counts due to shading effects.

## 2.4 Particle morphology determination

### 2.4.1 Projected-area and volume-equivalent diameters

In the present study, the image analysis integrated into the SEM-EDX software AZtec is used to determine the size of particles as a projected area diameter. Projected area diameter, $d_p$, is the diameter of a circle having the same area as the dust particle projected in a two-dimensional image and is calculated as:

$$d_p = \sqrt{\frac{4 \times A}{\pi}},$$ (1)

where $A$ is the area covered by the particle on the sample substrate.

Following Ott and Peters (2008), the volume-equivalent diameter (sphere with the same volume as an irregular shaped particle) also called the geometric diameter, $d_v$, is estimated from the projected area diameter via volumetric shape factor expressed by particle projected area and perimeter ($P$) as follows:





$$d_v = \frac{4\pi A}{P^2} d_p = \frac{1}{P^2}\sqrt{64\pi A^3}. \tag{2}$$

All particle diameters (d) presented here are converted from projected area-equivalent diameter to volume-equivalent (geo-metric) diameter (unless stated otherwise). The reason for this conversion is that geometric diameter is used in global aerosol models to quantify dust size (Mahowald et al., 2014) and optical properties depend on the particle volume.

### 2.4.2 Aspect Ratio

The two-dimensional (2D) shape of individual dust particles is presented here as aspect ratio (AR) and was calculated by the image analysis integrated into the SEM-EDX software AZtec. AZtec software manual defines AR as the ratio of the major to the minor axis of the elliptical fit on the projected particle area. Symmetrical features, such as spheres or cubes, have an AR that is approximately 1 whereas features that have shapes like ovals or needles have an AR that is greater than 1. A critical shortcoming of 2-D imaging is that it can yield different shapes of 3-D particles depending on their orientations on the sampling 200 substrate (Huang et al., 2020).

### 2.5 Number deposition rate calculation

The number deposition rate (NDR) is calculated from deposited particle numbers per area and individual particle size. First, a window correction ($c_w$) was applied (Kandler et al., 2009) to the particle deposition rate as

$$c_w = \frac{w_x w_y}{(w_x - d_p)(w_y - d_p)}, \tag{3}$$

where $w_x$ and $w_y$ are the dimensions of the analysis rectangle.

The NDR is then calculated as

$$NDR = \frac{1}{A \times t}\sum_i c_w(d_v, i), \tag{4}$$

where $A$ is the total analyzed area, $t$ is the sample collection time, and $i$ is the index of the particle.

### 2.5.1 Determining the size distributions from free-wing impactor measurements

Obtaining the atmospheric size distribution from FWI requires consideration of window correction and the collection efficiency dependence on the impaction speed and geometry. These corrections are applied to every individual particle as a function of its size and composition to get the overall collection efficiency ($c_e$) (for detailed formalism, see S2). The atmospheric concentration ($C$) is then computed from the deposition rate and velocity as



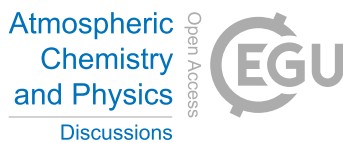

$$C = \frac{1}{A \times v \times t} \sum_i c_w(d_v, i) c_e, \tag{5}$$

where $A$ is the total analyzed area, $v$ is the impaction velocity, $t$ is the sample collection time, and $i$ is the index of the particle.

### 2.5.2    Determining the size distributions from the flat-plate sampler and cascade impactor measurements

Obtaining atmospheric concentrations from the NDR requires the use of different deposition velocity models for the flat-plate sampler. There exist a variety of models to estimate particle deposition speed based on a set of assumptions and neglections (Noll et al., 2001; Aluko and Noll, 2006; Piskunov, 2009; Petroff and Zhang, 2010). However, non of these models fitted the

observed deposition velocities, which appeared to be rather uniform in Waza et al. (2019). Therefore, for the present work, a constant deposition velocity for the size range of interest was assumed and the size distributions were normalized and averaged over all measurements. For the MOUDI samples, atmospheric concentrations cannot be determined with the analytic approach used in this work because the size of the impaction spot cannot be retrieved with high accuracy to get the total number of particles deposited at each spot. Therefore the same approach as FPS is applied to the MOUDI samples to get the size

distribution. In addition, direct comparison is often difficult among different instruments due to the different nature of particle collection from each technique (Chou et al., 2008; Price et al., 2018).

### 2.6    Particle classification and composition

SEM-EDX detects the elemental composition of individual dust particles. Some minerals have well-defined compositions and can therefore be identified relatively easy (e.g. gypsum, quartz, calcite). Others, e.g. clay minerals, are variable in nature,

so their identification is more ambiguous. Moreover, dust particles are commonly found in the form of internal mixtures or aggregates rather than in their pure mineral forms, and therefore the method used in this study identifies the major mineral type in each particle. Particle classification in this study is supported by the analysis of dust and soil samples for the major mineral composition from X-ray diffraction (XRD) measurements reported in a companion paper (González-Romero et al., in prep.). Furthermore, to assess artefacts, a few clean blank samples taken during the campaign were scanned and certain particle types

could be identified as contaminants and thus discarded from the analysis. The contaminants detected consisted of pure iron (apparently already from the manufacturing process) and F+Si. The contaminants were mostly rare compared to the abundance of dust particles.

Based on the single particle composition quantification, an elemental index for the element X is defined as the atomic ratio of the concentration of the element considered and the sum of the concentrations of the elements quantified (Kandler et al.,

240    2007, 2018)

$$|X| = \frac{X}{Na + Mg + Al + Si + P + S + Cl + K + Ca + Ti + Cr + Mn + Fe + Co}, \tag{6}$$



where the element symbols represent the relative contribution in atom % measured for each particle. Note that the given method cannot be used to quantify the percentage of C, N, and O due to their high uncertainty and substrate contributions. Classification is done using a set of rules, which use the elemental index and additional elemental ratios and are defined by the dominance of one or more specific elements or their ratios (e.g. Al, Si, Ca, Fe, Al/Si). Particle classes are named for their most prevalent component(s), which may contain terms for mineral phases to simplify the naming (e.g. gypsum, quartz). Those mineral phases were chosen as the most prevalent components that matched the reported elemental composition, but no real phase identification of individual particles (e.g., by transmission electron microscopy) was carried out. Therefore, all the particle classes are termed '-like' to express the similarity in chemical fingerprints. Details of the classification procedure are given in S1.

## 2.7 Ancillary data

Aerosol PSDs of suspended mineral dust were obtained by using an Optical Particle Counter (OPC, FIDAS 200, Palas GmbH, Germany). The measurement range of FIDAS was 0.2 - 19 $\mu$m with a time resolution of 2 min and a flow of 4.80 L min$^{-1}$. Data acquisition was performed by a data logger, averaging measurements over 2 min. In addition, meteorological data were acquired with a time resolution of 1 s. Air temperature and relative humidity were measured by HC2A-S3 sensor (Campbell Scientific) and wind speed, and wind direction was recorded by a two-dimensional sonic anemometer (Campbell Scientific WINDSONIC4-L) during the sampling period at approximately 0.4 m, 0.8 m, 2 m, 5 m and 10 m height facing north above ground level (see Fig. A1). The average OPC PSD is used here for comparison purposes. In a companion paper (González-Flórez et al., in prep.), the emitted PSD and its variability are analyzed in detail.

## 2.8 Statistical analysis

The uncertainty in the counting statistics for various particle groups was determined by generating confidence intervals assuming binomial distribution, i.e. for a relative number abundance of a particular particle group $r$, the two-sided 95 % confidence interval is approximated (Hartung et al., 2012). For the obtained PSDs, the only error source considered is the Poisson counting error.

## 3 Results and discussion

### 3.1 Observed particle types

Typical chemical compositions of a set of minerals and boundary rules were used to define particle groups (see S1 in the Supplement). Based on the applied scheme, particles were assigned to the following classes: (i) oxides/hydroxides, (ii) feldspar, (iii) clay minerals, (iv) quartz-like and complex quartz-like, (v) other silicates (vi) Ca-rich, (vii) sulfates, and (viii) mixtures and others. Figure 2 shows the average elemental composition of these groups. However, it is important to note that each particle





can be composed of different minerals and can have variable and even indistinguishable compositions. Therefore, the created groups may not only consist of the minerals described below.

### 3.1.1 Oxides/Hydroxides

This class consists of hematite-like and anatase-like particles with Fe and Ti X-ray signals dominating, respectively. Both particle types were mainly found in particles with $d_v < 2.5$ $\mu$m with anatase-like particles being rare. Iron (Fe) oxides are present both in pure crystalline form and as small impurities attached to other minerals. Moreover, Fe is present in other particle groups as well, mainly in clays and silicates. However, the presence of Fe can be due to not only Fe oxi(hydroxi)des but also to structural Fe. Fe oxides tend to exist as separate individual particles in sizes $d_v < 2$ $\mu$m and with increasing aggregate size could be distributed as small grains throughout the aggregate volume. The number abundance of this class was usually below 2 % in the collected samples.

### 3.1.2 Quartz-like and complex quartz-like

The classification of particles into quartz-like is relatively straightforward with a dominant Si signal. Some of the particles with high Si were also found with slightly higher Al content associated with them and are therefore classified as complex quartz-like. Fe content is generally low, suggesting that Fe oxides are not linked with quartz-like particles. The particle number abundance of quartz-like particles was $\sim 8$ % and complex quartz-like particles ranged from 1 % to 15 % depending on particle size.

### 3.1.3 Feldspar-like and pure feldspar abundance

The feldspar group is characterized by Si:Al:K / Si:Al:Na signal ratios of approximately 3:1:1 for microcline-like (K-feldspar) and albite-like (Na-feldspar), respectively. Ca-feldspar-like particles are quite rare in the samples. Na-feldspar particles are present across all size ranges ranging from 1.5 % in particles with $d_v < 2.5$ $\mu$m to around 3 % in particles with $d_v > 2.5$ $\mu$m. K-feldspar particles on the contrary are present in $<1$ % and are found mainly in particles with $d_v < 5$ $\mu$m. The complex feldspar-like and complex clay/feldspar mixture have Si:Al signal ratios of approximately 3:1 with additional elements such as Ca, Fe and Mg, suggesting a more complex aggregate than a pure feldspar. Complex feldspar-like particles were found in the range of 5 - 8 % in particles with $d_v < 20$ $\mu$m. For particles with $d_v > 20$ $\mu$m , its occurrence was about 15 %. Similar number abundances were also observed for complex clay/feldspar mixtures.

Regarding the identification of feldspars, there are two approaches: One is based on the classical 'classification' scheme, where a fixed limit around the usual composition of feldspars is used (detailed formalism in S1), and the other is the feldspar-specific index approach, where a distance from the ideal composition is calculated. Both form different shapes in the x-dimensional element space and therefore lead to different results. The index approach has the advantage that it shows how close a particle is to an ideal composition. In this case, the feldspar indices regard the overall contribution of feldspar-specific





**Figure 2.** Average elemental composition as a function of particle size for different particle groups. The legend shows element index for each respective element. The numbers on top represent total particle counts in the given size bin. Abundance bars are not shown for size bins with fewer than 10 particles.



elements to the particle and the specific Al/Si as well as alkali/Si or K/Si ratios. It was found that an index value of $> 0.80$ is suitable to distinguish between pure feldspar grains and other silicates. Details of the index calculation are given in S3.

### 3.1.4 Ca-rich

Calcium-rich particles include calcite-like, apatite-like, dolomite-like, and gypsum-like and these are characterized by high
Ca, Ca+P, Ca+Mg, and Ca+S content, respectively. In gypsum-like the Ca:S weight ratio is $\sim 1$ and in dolomite-like particles the Ca:Mg weight ratio is $\sim 1$. Apatite-like particles were quite rare and only present in a few samples. The majority of the particles were calcite-like and their abundance was usually around 8 % in the collected samples, with a higher proportion in $d_v$ $< 5$ $\mu$m. This group did not include particles with appreciable Fe contents in contrast to other classes (Fig. 2).

### 3.1.5 Clay minerals

Clay minerals can be divided into four subgroups: kaolinite-like, illite-like, smectite-like, and chlorite-like, with illite and kaolinite having a similar average Si-to-Al signal ratio (1.6 and 1.5 respectively), but with illite-like containing K, Fe and Mg whereas in kaolinite-like particles K is present in trace amount. These are the most abundant particle types found in the collected sample with number abundances of 25 % and 15 %, respectively. Smectite-like particles are clay minerals containing Fe and Mg, as well as small amounts of Ca and are found across all size ranges in small proportions ($< 1$ %). Chlorite-like
particles are characterized by slightly higher Fe content compared to other clay minerals and their relative contribution is around 5 % in particles with $d_v < 5$ $\mu$m and 2 % in particles with $d_v > 5$ $\mu$m.

### 3.1.6 Other silicates

The other silicate class group are characterized by the presence of Si and Al together with trace amounts of Mg, Ca, and Fe. Another silicate group in this class are Ca-rich silicates/Ca-Si-mixtures. These are typically clay minerals particles that are
internally mixed with Ca carbonates (e.g. calcite). Furthermore, Fe is also a minor component in this particle class suggesting an additional contribution of Fe oxides. The particle number abundance of other silicates class is $\sim 5$ % and that of Ca-Si mixtures varies between 5 % to 12 % depending on particle size.

### 3.1.7 Sulfates

In the sulfate group, mainly ammonium sulfate-like particles were present (i.e. only S was detected in a particle). They make
up a very small amount in the total sample ($< 0.25$ %) as individual particles and are mostly observed in particles with $d_v > 2$ $\mu$m. Ammonium sulfate is the prevalent sulfate species present in atmospheric aerosols and it has an anthropogenic secondary origin.





### 3.1.8 Mixtures and other

The silicate mixture group are dominated by Si along with the presence of other elements in varying amount and therefore does
not belong to any other defined silicate group. The so-called 'other' group contains all particles that are not classified by the
applied scheme into any of the groups described above. Therefore, it represents a mixture of different minerals or rare species.
The relative number abundance for mixtures was around 4 % in the collected sample and the number percentage of the 'other'
group is substantially low (< 0.5 %).

### 3.1.9 Silicate composition

The dominating composition in mineral dust is silicates which are composed of different minerals. Figure 3 shows the
highly variable chemical composition of the silicate particles with (Mg+Fe)/Si ratio (x-axis), Al/Si ratio (y-axis), and the
(Na+K+Ca)/Si ratio (z-axis, colour-coded) for each single particle. The three main clusters visible in Fig. 3 are numbered 1
(quartz-like), 2 (feldspar), and 3 (most other particles). Like mentioned in Kandler et al. (2011), the Al/Si ratio exhibits the
least measurement uncertainty and varies significantly for different minerals. The (Mg+Fe)/Si ratio is an indicator for clay
mineral aggregates as feldspar does not usually contain these elements (Anthony et al., 2003). The (Na+K+Ca)/Si ratio can be
used to differentiate between feldspar and clay minerals as feldspar shows much higher values as compared to clay minerals.
However, it can be also seen that this distinction is usually not clear-cut as all dust particles are usually 'contaminated' with
other minerals and the aggregates have ambiguous compositions.

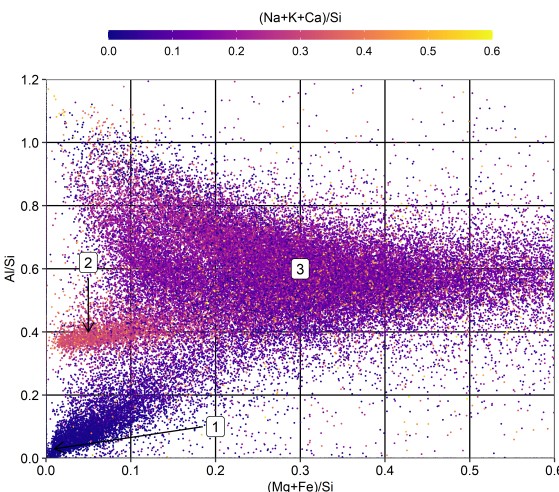

**Figure 3.** Scatter-plot for silicate particle composition. The three main clusters are marked by numbers: (1) quartz cluster, (2) feldspar cluster, and (3) clay mineral/aggregate cluster.





## 3.2 Comparison of PSD of different techniques for particle collection

Figure 4 (a) shows the normalized number size distributions obtained by FPS, MOUDI, and OPC. The size distribution obtained by FPS shows good agreement with that obtained from OPC. The size distribution of measured volume equivalent diameter is towards the higher end of the cut-off aerodynamic diameter expected on individual MOUDI stages. In stage 3, the mode is at around 3 $\mu$m (nominal cut point 2.5 $\mu$m) and for stage 4 the mode is at 1 $\mu$m (nominal cut point 1 $\mu$m). Figure 4 (b) shows the number size distribution obtained by FWI and is compared with the atmospheric size distribution from the OPC. A

clear discrepancy is observed between FWI and OPC which is larger than the statistical uncertainties. Here, FWI substantially underestimates particles < 10 $\mu$m. The collection efficiency of FWI is 50 % for 11 $\mu$m aerodynamic diameter particles and therefore for fine mode particles the efficiency correction function can result in unrealistic values. For diameter > 10 $\mu$m, the shape of the size distribution is similar to that of OPC but the observed concentrations are still an order of magnitude smaller. The discrepancy may be linked to one or more of the following potential sources of uncertainty: a badly defined efficiency curve

for FWI, the discrepancy in particle size definition (optical measured diameter versus projected area equivalent diameter) and particle density estimation from SEM to calculate volume equivalent diameter for efficiency correction. As the shape (and not absolute concentrations) of the number size distribution is quite similar among the different measurement techniques used and OPC and therefore provides confidence in the derived size-resolved elemental composition by SEM.

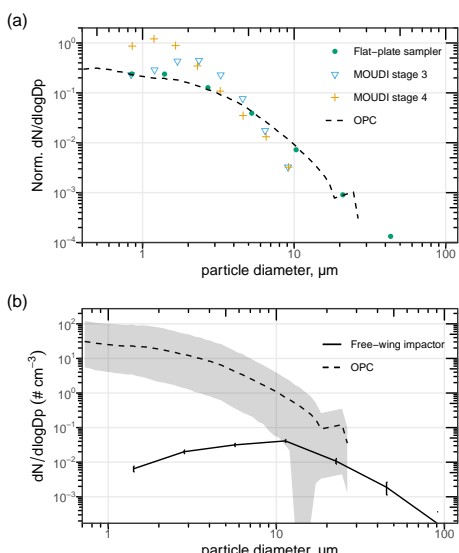

**Figure 4.** (a) Normalized number size distribution obtained by different sampling techniques averaged over the entire campaign period. (b) Comparing the atmospheric number size distribution between FWI and OPC. The bold line is for FWI with error bar showing the Poisson counting error and dashed line indicates the field campaign mean size distribution from OPC and the shaded region shows the 95th and 5th percentile. Note that below $\sim$ 15 $\mu$m diameter, the collection efficiency of the FWI drops significantly.





### 3.3 Particle composition

Figure 5 shows the time-averaged composition of the aerosol at L'Bour from samples collected with the three different sampling techniques. Since the composition of the aerosol particles in each sample does not exhibit a large sample-to-sample variability (see Fig. S1, S2, S3 for individual samples), all of the dust sample compositions were merged for further comparison. Relative number abundances are given in terms of particle group. A clear trend is seen among the different particle groups based on their size. Clay minerals are the most abundant and are present across all size ranges. Similarly, quartz-like particles are found

in each size class with almost similar number abundance (around 6 %) whereas the number fraction for complex quartz-like grows with particle size. This suggests that as particle size increases, quartz particles get internally mixed with elements like Al and Fe forming complex aggregates. A similar trend is also observed for feldspar particles where complex feldspar-like particles increase with particle size as a result of the inclusion of other elements creating a complex mixture. A detailed discussion on feldspar particles is addressed further in Sect. 3.3.4. Furthermore, the proportion of Ca-rich, Fe-rich, and sulfate-

like particles tends to decrease with increasing size. The Ca-Si mixture is slightly enhanced at $d_v > 4$ $\mu$m which could be due to the availability of more silicates to form aggregates with calcite and also the depletion of illite-like particles could partly form aggregates with calcite too.

On average across all particle sizes, 26 % of particles are illite-like, 14 % kaolinite-like, 10 % quartz and complex quartz-like, 7 % complex clay/feldspar mixture, 6 % Ca-rich silicate/Ca-silicate mixture, 5 % chlorite-like, 5 % calcite-like, 1.7 %

hematite-like, 1.7 % albite-like, 1.3 % gypsum-like, and 1.2 % smectite-like. The contribution of ammonium sulfate particles is restricted to particles with $d_v < 2.5$ $\mu$m. Chamber studies of Moroccan soil have shown a dominance of illite-smectite clay minerals in the size $< 2.5$ $\mu$m (Marsden et al., 2019). Our observed composition is fairly similar to the chemical composition of Saharan mineral dust collected during the SAMUM campaign from in situ measurements at Tinfou, Morocco in May/June 2006 (Kandler et al., 2009). However, a detailed comparison is not possible due to the inadequately subdivided silicates classes

in Kandler et al. (2009). Moreover, the observed composition type is corroborated with XRD analyses of particle size fractions separated from the sediments of L'Bour during the same sampling campaign (González-Romero et al., in prep.). Furthermore, samples from Morocco (Kandler et al., 2009) and transported Saharan dust at Tenerife (Alastuey et al., 2005) also showed similar composition from XRD analysis. Additionally, Kandler et al. (2009, 2011) found the fraction of quartz-like particles to increase with particle size. As dust originating from N Africa (Morocco, Algeria and Tunisia) can be subject to long-range

transport (Engelstaedter et al., 2006; Müller et al., 2009), events like red rains over Montseny mountains in Spain (Avila et al., 1997) and south-eastern part of Italy (Blanco et al., 2003) have been reported. Mineralogical analysis of samples from the studies above detected quartz, clay minerals (illite > kaolinite), and carbonates (calcite > dolomite) in a greater proportion with feldspar and smectite as other major mineral components (Avila et al., 1997; Alastuey et al., 2005). Few elements can be considered as a source tracer; for example, high Ca content is usually associated to be a tracer for dust emitted from Morocco

(Kandler et al., 2007) which is also found in our result by single particle analysis. Mineralogical database of soil composition also show high amounts of calcite and illite in north Sahara (Claquin et al., 1999) again corroborated by our single particle analysis.


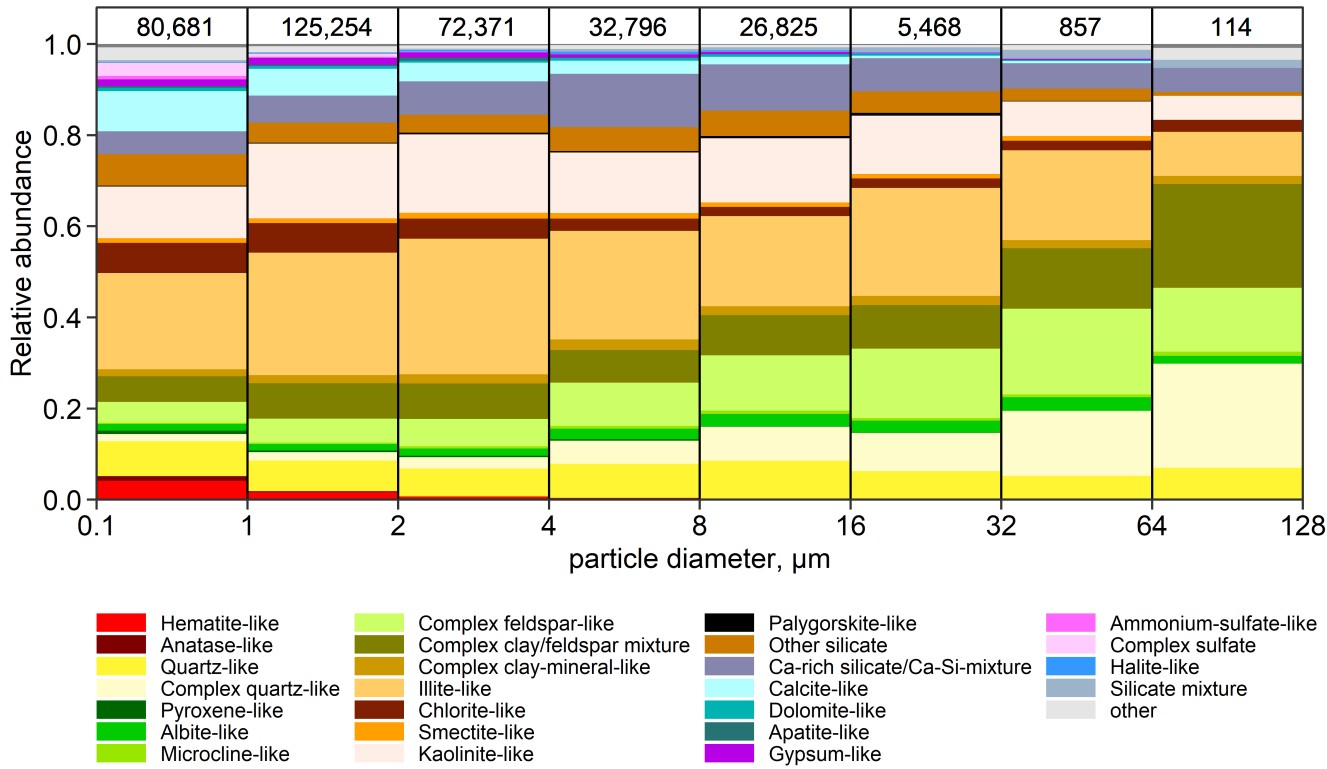

**Figure 5.** Average size-resolved relative number abundance of different particle groups. The numbers on top represent total particle counts in the given size bin.

### 3.3.1 Particle abundances using different techniques

Figure 6 shows the compositional information for dust samples obtained using 3 different collection techniques. The goal

was to get a representative number of particles across a wide size distribution which is usually the case for mineral dust. However, the collection of particles larger than a few micrometres is relatively difficult due to the poor collection efficiency of many instruments. Therefore, in this work an inlet-free impactor i.e. FWI was used to sample particles with $d_v > 3$ $\mu$m whereas FPS and MOUDI were used primarily for particles having $d_v < 3$ $\mu$m. In the MOUDI sample, most of the collected particles ($> 97\%$) had $d_v < 5$ $\mu$m. In stage 3 (cut-off $d_a$ 1 - 2.5 $\mu$m), 56 % of the collected particles were within the specified

stage size range; the size of half of the remaining particles was below the lower cutoff and that of the other half was above the upper cutoff. The major constituents were illite-like, kaolinite-like, quartz-like, Ca-silicate mixture, complex clay/feldspar mixture, and calcite-like with feldspar-like, quartz-like, hematite-like and sulfates as minor constituents. A similar trend was also observed in stage 4 of MOUDI except for an enhanced abundance of kaolinite-like and complex clay particles. In the FPS samples, the number of collected particles in the size range $0.1 < d_v < 5$ $\mu$m is quite comparable to stage 3 of MOUDI. For





particles with $d_v > 5\ \mu$m, the composition changes only slightly by increasing quartz-like and decreasing calcite-like particles. In the FWI samples, more than 70 % of particles had $d_v > 5\ \mu$m. The number abundance of feldspar-like, quartz-like, complex feldspar-like and complex clay/feldspar mixture particles increases with increasing particle size, while the number abundance of clay-like particles tends to decrease. In contrast, calcite-like, gypsum-like, and sulfates are virtually absent in samples from FWI in $d_v < 3\ \mu$m which is the size range where they are typically observed. However, the number of particles analysed in the

given size is significantly less compared to FPS or MOUDI. Note that it was not possible to perform sampling with FWI and MOUDI under the highest dust concentrations.

Differences in abundance for some of the classes among the different techniques might be due to the way by which particles reach the substrate. In MOUDI and FWI, particles are impacted on the substrate whereas in FPS particles are deposited by gravitational settling and turbulent diffusion. One hypothesis for such difference is that in the MOUDI the high impaction

speed experienced by particle aggregates on the top stages may break some of them into smaller ones and get carried away along the jet streams on the bottom stages. Such a hypothesis is consistent with the enhancement of clay-like aggregates in MOUDI. Other observed differences between the sampling techniques could be related to the non-parallel sampling times. MOUDI and FWI samples represent a few minutes compared to the usually half-day exposure time for FPS. However, as variations in the composition are fairly similar for all of our analysed samples, the latter is most likely not a major aspect.

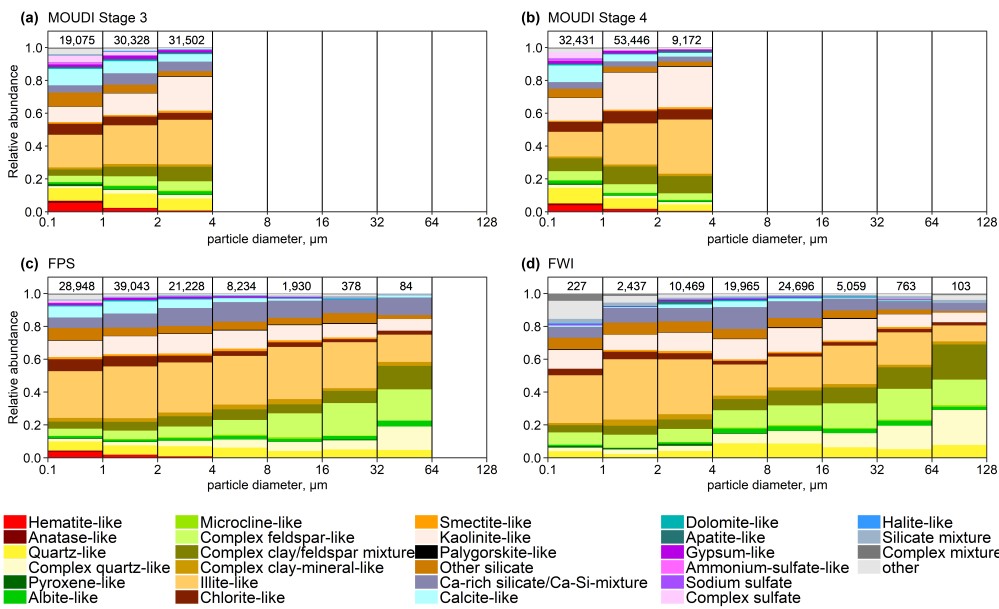

**Figure 6.** Average size-resolved relative number abundance of different particle groups using different sampling techniques collected over the campaign period. The numbers on top represent total particle counts in the given size bin. Abundance bars are not shown for size bins with fewer than 30 particles.





### 3.3.2 Temporal variability

Figure 7 displays time series of number abundances in three size ranges for the different particle groups observed in deposition samplers. While the dependence of chemical composition on particle size is quite strong, the temporal variation of the major particle groups does not show significant variability. This behaviour is to be expected as sampling was done in the source region, and the average daily composition is relatively constant.

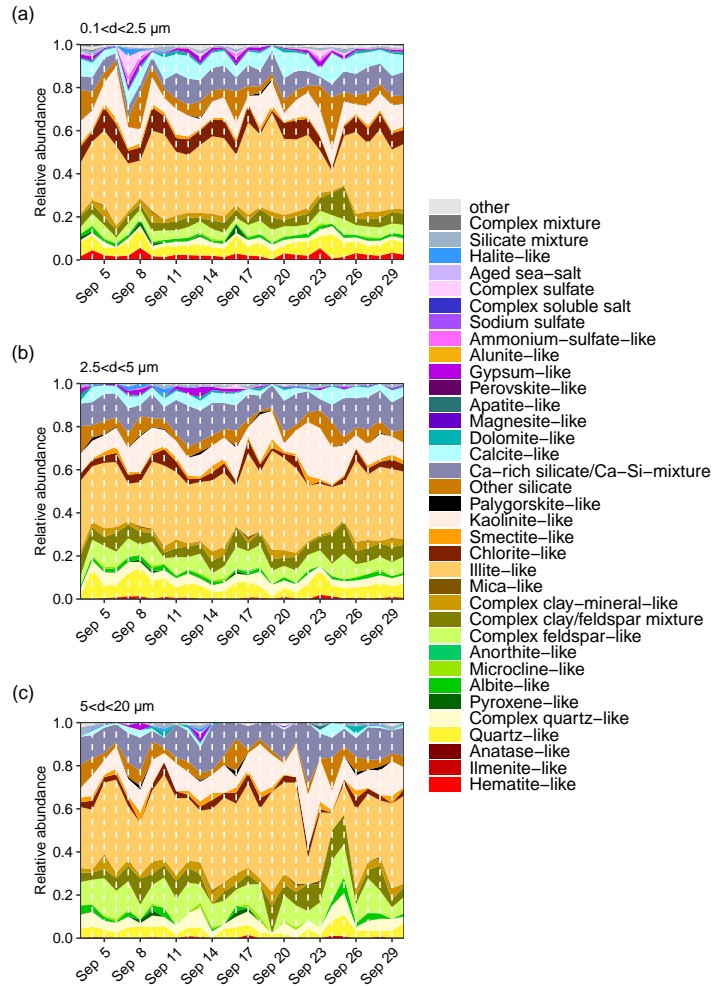

**Figure 7.** Time series of chemical composition for particles in (a) $0.1 < d < 2.5 \ \mu$m, (b) $2.5 < d < 5 \ \mu$m and (c) $5 < d < 20 \ \mu$m in L'Bour.

Nevertheless, looking at specific samples we can observe distinct compositional differences. Figure 8 shows chemical composition for four different situations. Secondary sulfates were dominant in particle $d_v < 2 \ \mu$m in samples collected during 8-10 September when the wind comes from the east (i.e. M'Hamid) suggesting anthropogenic origin which is further corroborated





with measurements of optical properties (Yus-Díez et al., in prep.) and PSD (González-Flórez et al., in prep.). This was also observed for small particles in the SAMUM campaign in Tinfou, Morocco (Kaaden et al., 2009). Apatite-like particles were

almost non-existent in our samples except for some specific days (20 and 23 September). As potassium (K) peak was not observed, windblown fertilizer as a source can be discarded. Furthermore, the particles were not beam-sensitive and no Cl and F signals were detected suggesting it to be OH-apatite originating from a distinct geogenic source. Most of these particles were confined to size $d_v$ 2-4 $\mu$m. In these particles, phosphorus is attached to the crystal lattice; which is one of the important dust-derived nutrients for marine and terrestrial ecosystems controlling phytoplankton productivity and carbon uptake (Stockdale

et al., 2016). The solubility of these particles is not known but can be estimated to be low (Prospero et al., 1996). Therefore, the amount of phosphorus available as a nutrient for marine or terrestrial ecosystems in our sample is expected to be relatively low. The sampling day (24 September) is characterized by elevated Ca-rich particles making it the most calcium-rich day and were present mainly in $d_v < 1$ $\mu$m. Here it was present in 30 % compared to the average of 5 %. This observation could be due to advection from a prominent dust source. Furthermore, the iron-rich particle is also quite pronounced on this day.

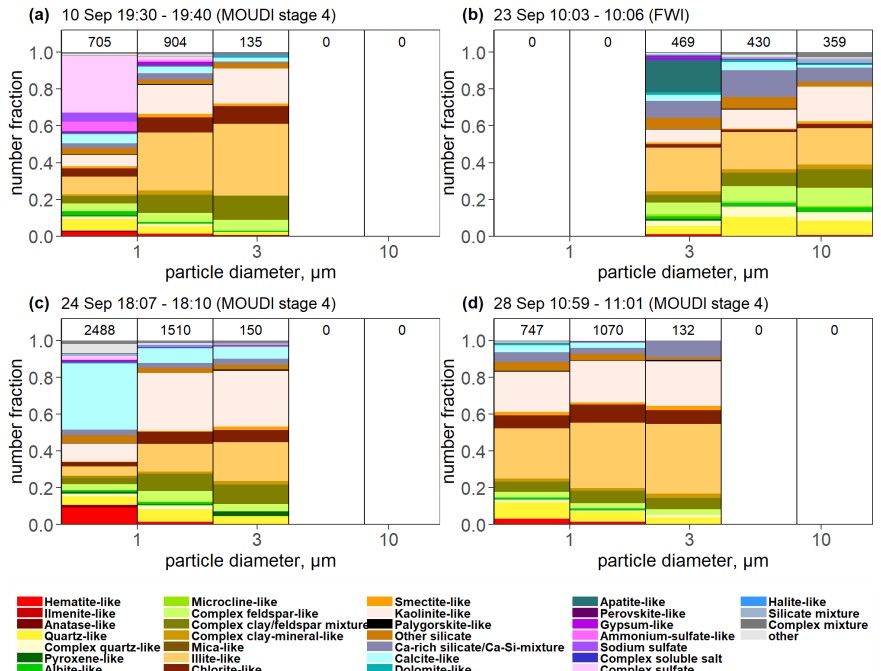

**Figure 8.** Size distribution of the particle relative number abundance for 4 different situations: (a) sulfate episode (b) apatite particle (c) Ca-rich episode and (d) typical average composition. The total number of analyzed particles in given for each size class with fewer than 30 particles not shown.





### 3.3.3  Iron distribution

Many minerals such as feldspars, clays and Fe-oxi/hydroxides contain Fe, which is a key ingredient to numerous climatic, environmental and biological processes affected by dust (Schulz et al., 2012). Fe in minerals can be broadly classified into structural Fe, which is typically found in the crystal lattice of clay minerals, and Fe in the form of oxide or hydroxide particles (mainly hematite and goethite) (Kandler et al., 2009; Scheuvens et al., 2011). The amount of free Fe-oxi/hydroxides along with its size, and aggregation or mixing state (internal vs external mixing) with other minerals determine the degree of absorption of solar radiation by dust (Sokolik and Toon, 1999; Moosmüller et al., 2012; Zhang et al., 2015; Di Biagio et al., 2019), and the potential Fe solubility (Baker and Jickells, 2006; Marcotte et al., 2020) of the deposited dust that fertilizes the ocean. In addition, it has been suggested that ocean primary productivity depends not only on the dissolved Fe but also on suspended solid Fe particles and their mineralogical components (Hettiarachchi et al., 2021). Furthermore, dust absorptive properties vary in different source regions mainly due to variations in the Fe oxides fraction (Lafon et al., 2006). While SEM cannot distinguish between structural and free Fe, by providing the Fe content on a particle-by-particle basis, it can provide some useful clues on the mixing state of Fe-oxi/hydroxides.

To understand the Fe content and to some extent the mixing state along with the type of particles that Fe is associated with, Fig. 9a shows the relative abundance of particles classified into nine categories according to their Fe content as a function of particle size, and Fig. 9b further displays the relative abundances of particles for each of the nine Fe categories and particle size ranges according to three Al/Si ratio categories. The Al/Si ratio is chosen because it has little measurement uncertainty, and varies significantly for different mineral groups as discussed in section 3.1.9. High Al/Si ratios indicate dominance of structural Fe-containing clay minerals, like illite, smectite and chlorite, which also tend to be internally mixed with Fe-oxi/hydroxides (Kandler et al., 2011). Low Al/Si ratios tend to be associated with quartz-like particles.

The relative number of particles with Fe fractions above 0.1 decreases with particle size. Among these, particles with Fe fractions above 0.5 are hematite-like particles as shown in Fig. 9a; their relative abundance is $\sim 4$ % below 1 micron, which decreases steeply with size to the extent that no appreciable amounts are observed above 4 microns. Given the amount of Fe, these hematite-like particles are mostly composed of Fe-oxi/hydroxides and to first order could be taken as externally mixed Fe-oxi/hydroxide particles. However, they still show some degree of aggregation with other minerals that increase with particle size. This can be appreciated in Fig. 2, where hematite-like particles show increasing amounts of Si, Al and Mg with size. Figure 9b shows that aggregation of Fe-oxi/hydroxides (Fe > 0.5) in hematite-like particles happens mainly with clays (high Al/Si ratios), but particles with low Al/Si ratios were also observed for diameters <8 $\mu$m. These are most likely for a fraction of particles (e.g. complex quartz-like) with some free Fe (oxy-hydr)oxides (e.g., goethite and hematite) assemblages.

Particles with Fe fractions between 0.2 and 0.5 follow a similar decrease in relative abundance with particle size although there are still appreciable amounts of particles up to 8 microns. Such particles are mostly classified as chlorite-like; in Fig. 2 chlorite particles feature Fe fractions above 0.2, and in Fig. 5 the abundance of chlorite-like particles decreases with particle size. However, the presence of Fe-oxi/hydroxide internal mixtures is very likely, particularly in particles with Fe contents towards the higher end of the 0.2-0.5 range. The relative number of particles with Fe fractions between 0.1 and 0.2 also





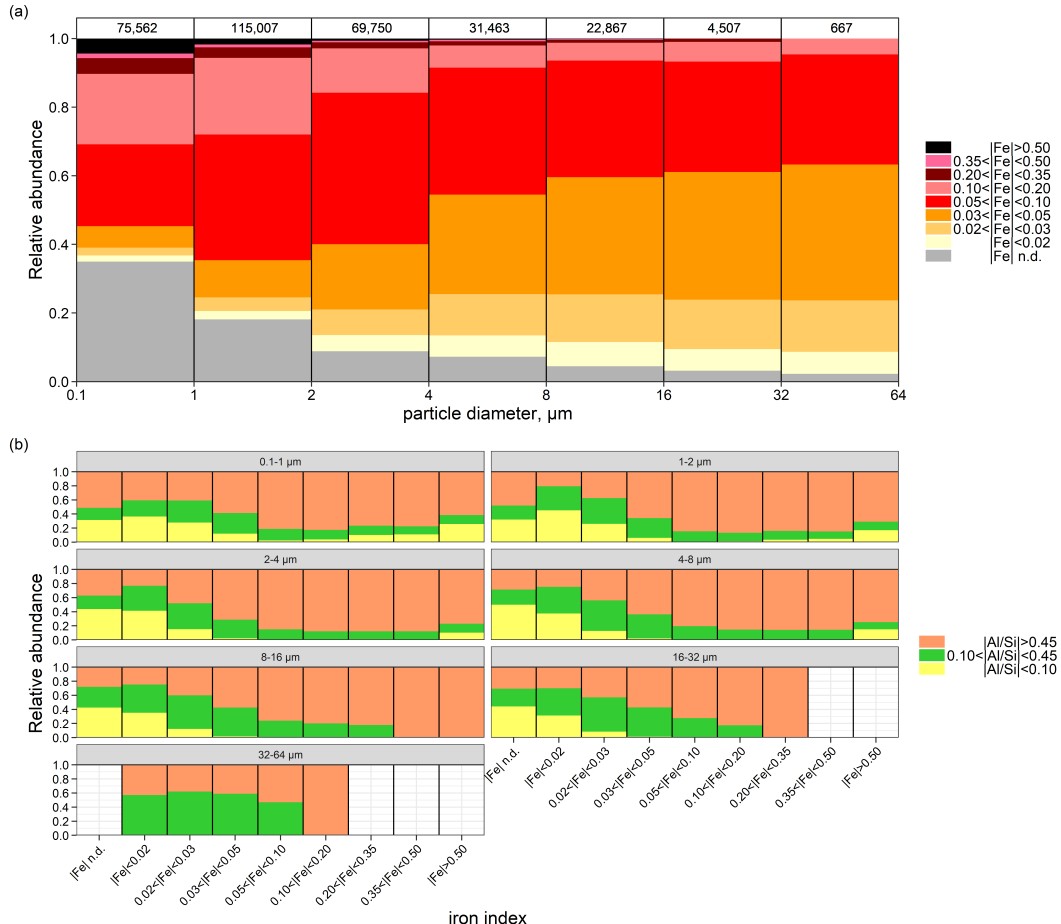

**Figure 9.** (a) Size resolved iron indices for silicate particles. "n.d." means Fe not detected. The numbers on top represent total particle counts in the given size bin. Abundance bars are not shown for size intervals with fewer than 40 particles because of high statistical uncertainty and (b) Silicate particle with iron association based on Al/Si ratio.

decreases with particle size; such particles have mainly a high Al/Si ratio (clay minerals, such as illite, smectite and chlorite) (Fig. 9b). Interestingly, the relative abundance of particles with no Fe detected reaches $\sim 0.38$ below 1 micron and thereafter also decreases with particle size. All in all, these results show that Fe-oxi/hydroxides tend to be increasingly internally mixed with other minerals, especially clays, as particle size increases. In other words, the volume fraction of Fe-oxi/hydroxides in aggregates decreases with particle size.

Conversely, with increasing particle size, the relative number of particles with Fe fractions between 0.03 - 0.05 increase, and that of particles with intermediate Fe content (0.05 - 0.10) tend to increase for particle sizes between 0.1 and 4 microns while remaining approximately invariant above $\sim 4$ microns. Such increasing trends with particle size are also visible for particles with Fe contents below 0.03. The increase in particles with lower Fe content with particle size is mostly due to the increasing





amount of complex clay-feldspar mixtures, complex feldspar-like and quartz-like aggregates (Fig. 5) with lower Al/Si ratios (Fig. 9b).

The mean Fe abundance is 0.09 and is dependent on size with values of 0.095 in the size range $0.1 < d < 2.5\ \mu$m and 0.06 for particles $2.5 < d < 64\ \mu$m. This is consistent with observations in Tinfou, Morocco (mean 0.053) for particles between 1 and 20 $\mu$m (Kandler et al., 2009). The Fe relative abundance in most of the particles varies between 0.03 and 0.1, suggesting that most of the Fe are built into the crystal lattice (structural) or internally mixed as small oxide/hydroxide grains (Kandler et al., 2009; Scheuvens et al., 2011; Deboudt et al., 2012; Kandler et al., 2020).

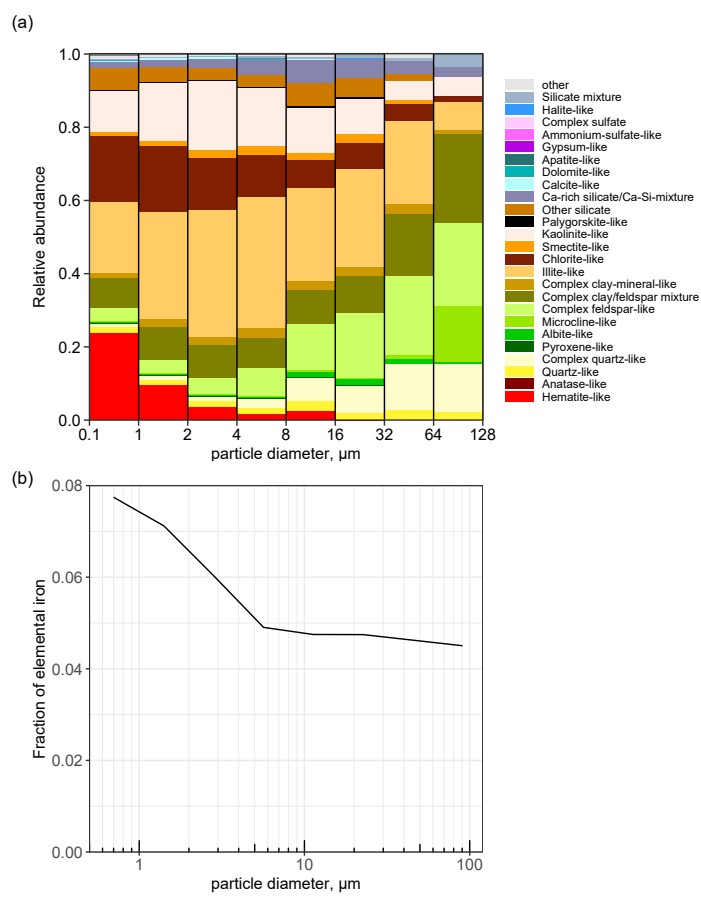

**Figure 10.** (a) Relative contribution of each particle type to total iron and (b) fraction of elemental iron by mass with respect to all the other elements.

Figure 10a shows the relative contribution of each particle type to Fe in every particle class. In comparison to the particle type fractions displayed in Fig. 5, the hematite-like and chlorite-like particles have an increased proportion because of their higher contribution in terms of Fe content, and the calcite-like and Ca-rich silicate group gets substantially diminished due to the absence of Fe in carbonate minerals. It is remarkable to see that the iron contribution of clay-like aggregates (illite, smectite





and kaolinite) increases with particle size up to $\sim$ 4-8 $\mu$m, which suggests (and further confirms) increased internal mixing
with Fe-oxi-hydroxides. Figure 10b shows the mass fraction of elemental Fe as a function of particle size. The total Fe in
dust is greatly reduced with size, particularly up to $\sim$ 5 $\mu$m, going from $\sim$ 8 % below 1 micron to less than 5 % in particles
above 6 $\mu$m. This is slightly higher than the values of fractional mass of elemental iron obtained for Moroccan soil using X-ray
Fluorescence (XRF) analysis which was reported to be 3.6 % for PM$_{10.6}$ (Caponi et al., 2017; Di Biagio et al., 2019).

Overall, our results suggest that smaller particles are more enriched with Fe due to both the presence of small-sized Fe-oxi-
hydroxide (hematite-like) and the increase in feldspar and quartz with particle size, which reduces the fractional abundance
of clay-like particles internally mixed with Fe-oxi-hydroxides. As fine dust particles are more prone to long-range transport,
this might provide an increased amount of soluble Fe. However, due to the association of Fe in different particle groups, it
is difficult to estimate the fraction of soluble Fe available after deposition in ocean biogeochemistry and thereby the global
carbon cycle or terrestrial ecosystem as a nutrient with the method used in this work. Nevertheless, as observed by Marcotte
et al. (2020), the Fe solubility (% dissolved Fe/total Fe in the mineral) in pure Fe-bearing clay materials was higher by an order
of magnitude than in pure Fe oxide minerals so the dust transported from this region could be a major source nutrient.

### 3.3.4   Feldspar

Feldspar, and in particular K-rich feldspar, are discussed as the most efficient ice nuclei among the different mineral constituents
found in dust (Atkinson et al., 2013; Harrison et al., 2016; DeMott et al., 2018) and are therefore of interest for atmospheric
processes.

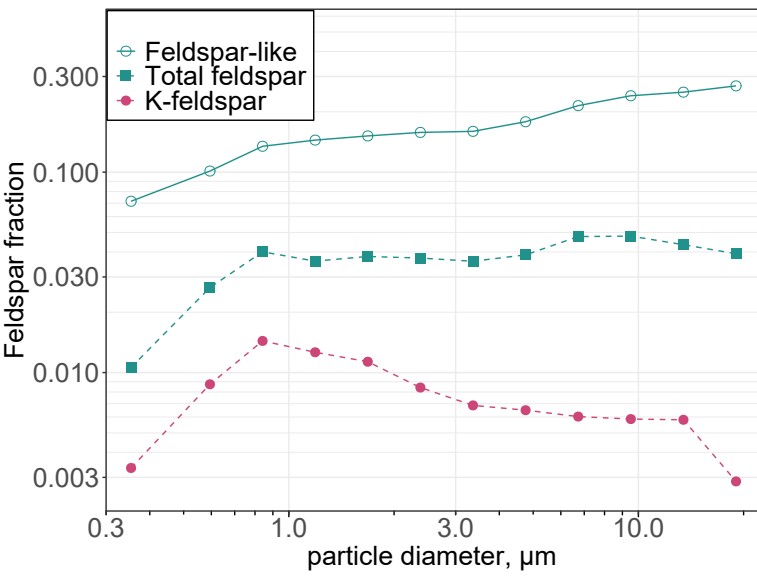

**Figure 11.** Number ratio of Feldspar-like, Total feldspar (more strict criterion), and K-feldspar (more strict criterion) to total particles.





Figure 11 shows the feldspar relative number abundance for K-feldspar and total feldspar to total particles as determined by the feldspar index approach. Here the total feldspar includes K-feldspar, Na-feldspar, and other feldspar. For comparison, the 'classical' classification scheme of feldspar-like particles is also shown which includes albite-like, microcline-like, anorthite-like, complex feldspar-like, and complex clay/feldspar mixture. In general, our analysis shows that approximately 4 % of the

dust aerosols are pure externally mixed feldspar particles whereas pure externally mixed K-feldspar particles are less than 1 %. With the inclusion of internally mixed feldspar, an increasing trend is observed with particle size. Studies have found the K-feldspar content of dust to be highly variable with values up to 25 % for Moroccan dust (Atkinson et al., 2013). Since the size distribution of K-feldspar at the source determines how much of it is transported across long distances, our finding shows that the relative amount of externally mixed K-feldspar particle is size independent up to 10 $\mu$m. For long-range transported

dust over Barbados, no significant variation was reported for K-feldspar under different dust conditions but are slightly lower than the values reported here using the same technique as in this study (Kandler et al., 2018; Harrison et al., 2022).

### 3.4 Particle shape

Dust particles were found to be highly aspherical and displayed a wide range of AR values for the size range $0.5 < D_p < 100$ $\mu$m, which is consistent with previous studies of Northern African dust (Chou et al., 2008; Kandler et al., 2009). The

median AR of particles is 1.46 and is almost independent of particle diameter as seen in Fig. 12 (b). This is consistent with the median AR (1.50) obtained by the analysis of Aerosol Robotic Network Sun photometer for dominant particles in desert dust plumes (Dubovik et al., 2006). It is however slightly lower than the one found in previous AR measurements of Moroccan dust (median 1.6) (Kandler et al., 2009) but is similar to the Asian dust (median 1.4) (Okada et al., 2001) and those observed in Niger during aircraft campaign (Matsuki et al., 2010). These are also substantially lower than values reported for long-range

transported Saharan dust (Reid et al., 2003; Coz et al., 2009). The discrepancy in observations at the point of the source region and transported dust could be the preferential removal of lower AR particles at the source due to their higher gravitational settling velocity than elongated particles of the same volume (Yang et al., 2013). Other median values found in the literature at the source location are 1.7 during African Monsoon Multidisciplinary Analysis (AMMA) campaign (Chou et al., 2008), 1.55 during Geostationary Earth Radiation Budget Intercomparisons of Longwave and Shortwave radiation (GERBILS) campaign

(Klaver et al., 2011). Median value of 1.3 was found during the Fennec campaign (Rocha-Lima et al., 2018) for particles < 5 $\mu$m and during AERosol Properties - Dust (AER-D) aircraft measurement median of 1.3-1.44 for 0.5 to 5 $\mu$m, 1.30 for 5 to 10 $\mu$m and 1.51 for 10 to 40 $\mu$m was observed (Ryder et al., 2018).

It should be noted that the AR calculation done in several of the studies mentioned above are not directly comparable as systematic difference exists due to non-uniformity in the definition of the aspect ratio as well as the software algorithms used in

computing AR (Huang et al., 2020). For example the median AR of Asian dust stated as 1.4 in Okada et al. (2001) transforms to 1.68 depending on the way length and width of individual dust particles are defined (Huang et al., 2020).

The density distribution with respect to the AR can be parameterized by a modified log-normal distribution (Kandler et al., 2007):





(a)

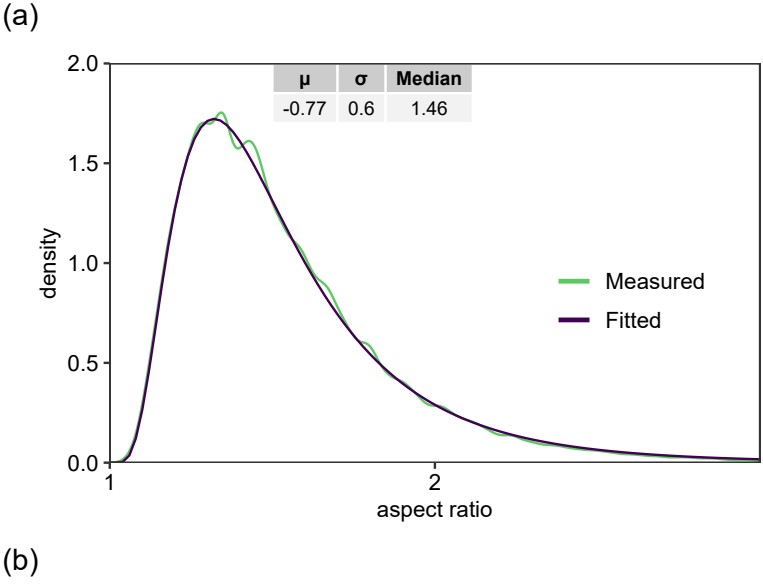

(b)

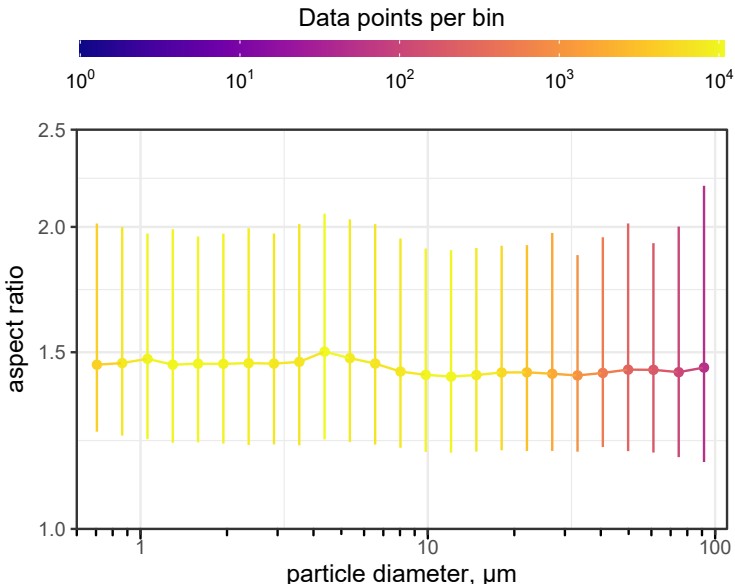

**Figure 12.** (a) Measured (green) and parameterized (purple) aspect ratio density distribution of dust particles (b) Size-resolved particle aspect ratio. The shaded area represents the range between 0.1 and 0.9 quantiles with dot being the median and the bins are color coded by the number of data points within each bin.





**Table 1.** Parameters of the aspect ratio density distribution, mean, median, and standard deviation for different particle classes

| Particle type | $\sigma$ | $\mu$ | Mean | Median | St. Dev. |
|---|---|---|---|---|---|
| Hematite-like | 0.5754 | -0.7942 | 1.51 | 1.44 | 0.30 |
| Quartz-like | 0.5917 | -0.7965 | 1.54 | 1.45 | 0.40 |
| Complex quartz-like | 0.5991 | -0.7793 | 1.54 | 1.45 | 0.44 |
| Microcline-like | 0.5701 | -0.7263 | 1.53 | 1.46 | 0.30 |
| Albite-like | 0.5852 | -0.7493 | 1.53 | 1.46 | 0.33 |
| Complex feldspar-like | 0.6204 | -0.7894 | 1.54 | 1.45 | 0.38 |
| Complex clay/feldspar mixture | 0.5990 | -0.8248 | 1.52 | 1.43 | 0.34 |
| Complex clay-mineral-like | 0.6008 | -0.7078 | 1.57 | 1.49 | 0.35 |
| Illite-like | 0.5936 | -0.7500 | 1.55 | 1.47 | 0.34 |
| Chlorite-like | 0.5928 | -0.7268 | 1.55 | 1.47 | 0.34 |
| Smectite-like | 0.6113 | -0.7390 | 1.55 | 1.46 | 0.35 |
| Kaolinite-like | 0.6056 | -0.8136 | 1.52 | 1.44 | 0.35 |
| Ca-rich silicate/Ca-Si-mixture | 0.6183 | -0.7006 | 1.58 | 1.50 | 0.40 |
| Calcite-like | 0.5904 | -0.7572 | 1.55 | 1.46 | 0.37 |
| Dolomite-like | 0.5931 | -0.9837 | 1.44 | 1.35 | 0.30 |
| Gypsum-like | 0.6505 | -0.7287 | 1.60 | 1.46 | 0.49 |
| **All** | 0.6024 | -0.7744 | 1.55 | 1.46 | 0.38 |

$$h(AR) = \frac{1}{\sqrt{2\pi} \cdot (AR-1) \cdot \sigma} \times exp\left[ -\frac{1}{2}\left( \frac{ln(AR-1)-\mu}{\sigma} \right)^2 \right] \qquad (7)$$

The fitting parameters $\sigma$ and $\mu$ along with the median value are given in Fig. 12 (a) for all particles and in table 1 for different particle classes. Most of the particle groups have a similar median AR of 1.46 except for dolomite which has the lowest median AR of 1.35. While the AR is generally independent of particle size and particle types, slightly higher aspect ratios are observed for internally mixed particles. Differences with other locations could then be at least partly explained by differences in mineralogical composition and mixing state.

## 550   4   Conclusions

We investigated the single particle composition of freshly emitted mineral dust aerosols using SEM-EDX. Samples were collected between 4-30 September 2019 during field campaign at a major source region on the edge of the Saharan desert in





Morocco (29.83 °N 5.87 °W) in the context of the FRAGMENT project. A total of more than 300,000 individual particles were classified into eight major particle classes based on their elemental composition and were: (i) oxides/hydroxides, (ii) feldspar, 555 (iii) clay minerals, (iv) quartz-like and complex quartz-like, (v) other silicates (vi) Ca-rich, (vii) sulfates, and (viii) mixtures and others. Our analysis shows that clay minerals (illite-like, kaolinite-like) were the most abundant across the size range $d_v < 64$ $\mu$m (28 %–48 % in relative number abundance). In contrast, calcite-like and chlorite-like particles were present mainly in the size range $d_v < 4$ $\mu$m, (4 %–9 %). Quartz-like particles were present in a relatively constant quantity across different size bins (5 % –10 %) whereas complex quartz-like particles were present primarily in the size range $d_v > 4$ $\mu$m, (4 %–9 %). A similarly 560 high fraction of quartz, feldspars, and clay minerals were reported for Morocco during SAMUM (Kandler et al., 2009), except for Ca-rich particles which are significantly less in our sample whereas in Kandler et al. (2009) appear as a log-normal mode centred around 3 $\mu$m. Particles with high iron content (Fe oxides and hydroxides) were also identified which are crucial for assessing the radiative effect of mineral dust as well as are essential nutrients for marine and terrestrial ecosystems.

The time series of the overall composition of mineral dust showed relative homogeneity over the campaign. Nevertheless, 565 elevated contents of apatite, calcite, and sulfates were observed on certain days suggesting the presence of a strong local source.

Most particle groups have median aspect ratios of around 1.46. Slightly higher aspect ratios are observed for internally-mixed particles whereas dolomite-like particles have the lowest median value of 1.35. This implies that dust aggregates with higher internal mixture fractions tend to have higher aspect ratios, which will change the radiative properties of dust. Studies have shown a trend of increasing aspect ratios for Northern African dust after transatlantic advection (Huang et al., 2020).

We provided a more exhaustive analysis of Fe-oxi/hydroxides and feldspar, including their size distribution and potential aggregation or mixing state, which are key to the effects of dust upon radiation and clouds. Compared to clay minerals and quartz, iron-rich particles (either dominated by or externally-mixed Fe-oxi/hydroxides) are present mainly in the size range $d_v < 1$ $\mu$m (4 % number abundance). Our analysis suggests that Fe-oxi/hydroxides tend to be increasingly internally mixed with other minerals, especially clays, as particle size increases, with the volume fraction of Fe-oxi/hydroxides in aggregates 575 decreasing with particle size. All in all, smaller particles are more enriched with Fe due to both the presence of Fe-oxi-hydroxide (hematite-like) and the increase in feldspar and quartz with particle size, which reduces the fractional abundance of clay-like particles internally mixed with Fe-oxi-hydroxides. The externally-mixed total feldspar and K-feldspar abundances are relatively constant in contrast to the increasing abundance of feldspar-like (internally-mixed) aggregates with particle size.

Many earth system models still assume globally uniform mineralogical composition and therefore introduce errors in the 580 assessment of regional forcing of dust. The detailed single-particle analysis of freshly emitted dust particles in this study can help include size-resolved composition dependencies in climate models. More recent FRAGMENT field campaigns in Iceland and Jordan will supplement these measurements in understanding the source-specific mineralogical composition of dust aerosols. Recently, the field of imaging spectrometry satellite missions has progressed with the development of new instruments and projects like Earth Surface Mineral Dust Source Investigation (EMIT) (Green et al., 2020), Environmental 585 Mapping and Analysis Program (EnMAP) (Guanter et al., 2015), and DLR Earth Sensing Imaging Spectrometer (DESIS) (Müller et al., 2016) that will retrieve global surface mineralogy. Together, all these efforts will help advance our knowledge of dust effects upon climate.





*Data availability.* The data presented in this work will be made available at an online repository and will be linked with a doi.

**Appendix A**

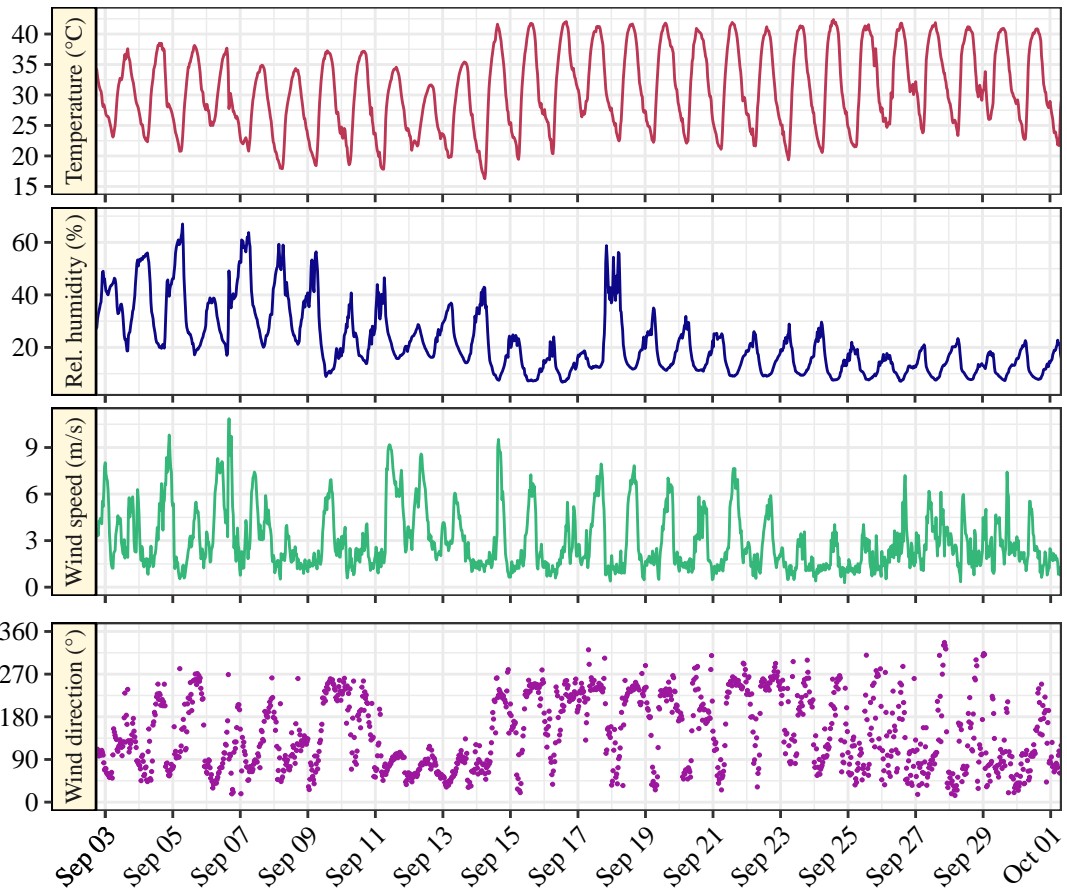

**Figure A1.** Time series of temperature, relative humidity, wind speed, and wind direction at approximately 2 m height. A 30 min average is applied on all time series presented.

*Author contributions.* CPG-P proposed and designed the measurement campaign with contributions from AA, KK, MK, and XQ. AP, KK, AA, CGF, AGR, MK, XQ, CR, JYD, and CPG-P implemented the field campaign. AP collected the samples, analysed them by electron microscopy, performed formal analysis, visualization (except Fig. 1 (a), and (b)), and writing of the original draft manuscript. KK and CPG-



P supervised the work. AGR generated Fig. 1(a) and MK generated Fig. 1(b). KK programmed data processing code and interpretation. CPG-P re-edited the manuscript. All authors contributed in data discussion and manuscript finalization.

*Competing interests.*   The authors declare that they have no conflict of interest.

*Acknowledgements.*   The field campaign and its associated research, including this work, was primarily funded by the European Research Council under the Horizon 2020 research and innovation programme through the ERC Consolidator Grant FRAGMENT (grant agreement No. 773051) and the AXA Research Fund through the AXA Chair on Sand and Dust Storms at BSC. CGF was supported by a PhD fellowship from the Agència de Gestió d'Ajuts Universitaris i de Recerca (AGAUR) grant 2020_FI B 00678. KK was funded by the Deutsche

Forschungsgemeinschaft (DFG, German Research Foundation) – 264907654; 416816480. MK received funding through the Helmholtz Association's Initiative and Networking Fund (grant agreement no. VH-NG-1533).

We acknowledge the EMIT project, which is supported by the NASA Earth Venture Instrument program, under the Earth Science Division of the Science Mission Directorate. We thank Dr. Santiago Beguería from the National Scientific Council of Spain for facilitating a field site in Zaragoza, Spain, to test our instrumentaltion and field procedures prior to our campaign in Morocco. We thank Prof. Kamal Taj Eddine

from Cady Ayyad University, Marrakesh, Morocco for his invaluable support and suggestions for the preparation of the field campaign. We thank Houssine Dakhamat and the crew of Hotel Chez le Pacha in M'hamid El Ghizlane for their support during the campaign.



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
