# Peer review of "Insights into the single particle composition, size, mixing state and aspect ratio of freshly emitted mineral dust from field measurements in the Moroccan Sahara using electron microscopy"

_Atmospheric Chemistry and Physics, 2022_

## Referee Comment (RC1)

**Insights into the single particle composition, size, mixing state and aspect ratio of freshly emitted mineral dust from field measurements in the Moroccan Sahara using electron microscopy**

Agnesh Panta, Konrad Kandler, Andres Alastuey, Cristina González-Flórez, Adolfo González-Romero, Martina Klose, Xavier Querol, Cristina Reche, Jesús Yus-Díez, and Carlos Pérez García-Pando

**General Comments**

The manuscript describes the sampling and analysis of mineral dust samples collected in the Moroccan Sahara in September 2019. For inter-comparison the dust particles were collected by three sampling methods, including flat-plate sampler (FPS), free-wing impactor (FWI), and a multi-stage deposition impactor (MOUDI). Real time particle size distributions were also measured by optical particle counter (OPC). The samples were analyzed by automated scanning electron microscopy (SEM), generating analyses on particle morphology and chemical composition, of more than 300,000 individual dust particles. Normalized elemental results were assembled into eight particle classes by a set of rules, each class approximating a likely dust mineral. These were labelled as hematite-like, quartz-like, feldspar-like, calcite-like, gypsum-like, halite-like ammonium sulphate-like, kaolinite-like, etc. The results were classified by particle diameter into eight size bins from, < 1 to 128 μm, each size bin containing the particle counts in that size bin. Inter-comparative plots of number abundances of the mineral-like and other chemical parameters abundances are provided. Discussion of the iron comparative number distribution plots for the FPS, FWI, two stages of the MOUDI, as well as for the OPC are given, with discussion of the mixing state thereof. Aspect ratios are compared with those from other studies.

The title clearly reflects the content of the paper and the abstract provides a concise summary. It adequately addresses the *in situ* sampling and SEM analysis of mineral dusts, of importance to a better understanding of airborne minerals and their potential impact on Global climate.

Although automated SEM analysis has been known for more than two decades (Engelbrecht et al., 2009), the application thereof to dust analysis remains novel, especially regarding the interpretation of data. Research

presented in this paper further explores the automated SEM procedures and technology. A Global library of

30  airborne dust mineralogical and morphological data will facilitate the interpretation of satellite signals.  To

readily compare with other similar studies, the chemical results should be calculated to reflect mass fractions or

mass percentages, not number abundances.

**Specific Comments**

35     1.  Particle **numbe**r abundances (fractions) per "volume equivalent diameter" are presented in figures and

text in the manuscript.  In a few instances (lines 16, 447, 532, 574) the authors do refer to volume

fractions. As shown in the below figures (Engelbrecht et al., 2017), there is a substantial bias towards

the number of smaller particles. To better compare with other bulk analytical techniques, individual

particle **volume** or **mass** (Ott and Peters, 2008a; Prakash et al., 2016; Engelbrecht et al., 2009;

40     Engelbrecht et al., 2017; Marsden et al., 2018; Marsden et al., 2019; Ott and Peters, 2008b) can be

assessed from SEM based particle area measurements (Engelbrecht et al., 2009; Prakash et al., 2016;

Ott and Peters, 2008b).  Similarly, for intercomparison of chemical parameters between particle size

bins, it is preferable to express abundances as equivalent **volumes**, and if particle densities are known,

particle mass.

45         It is recommended that the number concentrations be converted to at least volume concentrations

throughout this manuscript, and that the chemical parameters be assessed in terms of volume fractions

or volume percentages. The figures illustrating the chemical parameters will need to be re-compiled to

reflect volume (or mass) concentrations and not number abundances. In this way the results can be

readily compared to other SEM based studies, as well as to chemical results generated by other

50     analytical techniques such as SPMS, ICP-MS, XRF, etc.

[Figure]

[Figure]

**Dust particle size distribution by SEM**

Number concentration

Volume concentration

[Figure]

[Figure]

Particle size distribution of sample collected by inverted "Frisbee" sampler on KAUST campus in December 2015, by (a) number, and by (b) volume, of about 2,000 particles measured by scanning electron microscopy (SEM)

2. Results from this study should be discussed in conjunction with individual particle dust studies from other desert regions, including North Africa and the Middle East (Engelbrecht et al., 2009; Engelbrecht et al., 2017; Prakash et al., 2016).

**Some Technical Corrections to consider**

Line 43. properties and mineral composition

Line 47.  deposit faster due to gravity, ….

Line 53.  ..dust is more optically absorbing when…

Line 57.  ..not only on the source composition but also on the feldspar…

Line 80.  …are the interdependencies among particle size, composition,…

Line 82.  …mixing state upon both particle composition and size…

Line 83. …mineralogical composition of windblown dust are relatively new …

Line 89.  Unclear, rephrase sentence. "One reason for such measurement scarcity is that source areas, particularly the host prolific ones, are typically located in remote areas and subject to harsh conditions."

Line 91. Dust storms result in high particulate concentrations and filter overload, providing a challenge to analyze by automated SEM.

Line 96. Three different instruments were used to sample mineral dusts, for the measurement of chemical and

70 physical properties of individual dust particles by SEM coupled with an energy-dispersive X-ray analyzer (EDX).

Line 99. …individual dust minerals to ultimately help understand…

Line 101. Such information is required to better constrain climate models that consider mineralogical variations in their representation of the dust cycle (Perlwitz et al., 2015a; Scanza et al., 2015; Li et al., 2021). This is timely, given the prospect of global soil mineralogy retrievals using high-quality satellite hyperspectral measurements

75 (Green et al., 2020).

Line 117. Under favorable weather conditions, dust is frequently emitted from this source area. (González-Flórez et al., in prep.).

Line 129. …FWI were collected twice daily with a typical….

Line 138. …particle size cutoff is determined by the impaction …

80 Line 145. The main controls for particle …

Line 153. …and fourth stages were selected for detailed analysis.

Line 154. ..when particles impact on the collection substrate…..

Line 156. ….depending on the mineralogical properties of particles….

Line 162. Previous studies (Kandler et al., 2008………

85 Line 162. Add references (Engelbrecht et al., 2009; Engelbrecht et al., 2017; Prakash et al., 2016)

Line 166. ….without any pretreatment. Particles were automatically detected and analyzed …….

Line 167. …(BSE) imagery was used for particle detection, as dust particles ….

Line 170. .. number of input counts by the EDX detector.

Line 174. Accounting for matrix dependent efficiencies of the….

90 Line 176. The SEM-EDX results are normalized to 100 %, including those of the elements C, N, O, Na, Mg, Al, Si, P, S, Cl, K, Ca, V, Cr, Mn, Fe, Zn, and Pb. …….

Commnent: What is the minimum particle size measured by automated SEM-EDX?

Line 228. SEM-EDX measures the elemental composition

Line 229. …can therefore be readily identified (e.g. gypsum, quartz, calcite).

95    Line 238. X is defined as the ratio of the mass concentration of the element considered to the sum of the mass concentrations of all the measured elements.

Line 243. …method was not applied to quantify…

Line 246. Their most prevalent chemical components, …

Line 247. The mineral labels were assigned from the most prevalent measured elemental concentrations. There

100   was no actual phase identification ……

Line 252. Aerosol PSDs of suspended mineral dusts were also obtained by Optical Particle Counter (OPC)…….

Line 264. …..only error considered is the Poisson counting error. Explain or delete sentence

Line 268   Based on set classification rules, particles….

Line 275 …particle types were found mainly in particles with…

105   Line 284. ….Fe content of quartz-like particles is generally low and variable, suggesting that Fe oxides are not an integral part of the quartz-like particles.

Line 289. Ca-feldspar-like particles are quite rare in the samples. Explain why, plagioclases are seldom pure albite, often oligoclase or andesine, depending on their provenance. This anomaly perhaps to do with the SEM analytical procedure?

110   Line 301. Figure 2 y-axis. Are relative abundances by number or volume (or mass)?

Line 326. Ammonium sulfate is the prevalent sulfate species in atmospheric aerosols and generally of anthropogenic origin.

Line 373. On average across all particle sizes, by number, 26% of particles are illite-like, ……

Line 416. …. along the air stream onto the lower stages.

115   Line 422. While the dependence of particle size distribution on sample mineralogy is quite strong, the temporal …

Line 431. There are fertilizers that do not contain K, such as di-ammonium phosphate (DAP), mono-ammonium phosphate (MAP). Hydroxy apatite is not commonly found in the Morocco phosphate deposits, the mined francolite generally contains F.

120   Line 439. Figure 8. Phase compositions of size bins for samples collected under specific meteorological conditions at specific times and on certain days.

Line 439. The total number of analyzed particles is given for…..

Line 441. feldspar not a typical Fe bearing mineral, only K-feldspar in very small amounts.

Line 474, Figure 9. Relative number abundances?

125

Engelbrecht, J. P., McDonald, E. V., Gillies, J. A., Jayanty, R. K. M., Casuccio, G., and Gertler, A. W.: Characterizing mineral dusts and other aerosols from the Middle East – Part 1: Ambient sampling, Inhalation Toxicology, 21, 4, 297-326, doi: 10.1080/08958370802464273, 2009.

130 Engelbrecht, J. P., Stenchikov, G., Prakash, P. J., Lersch, T., Anisimov, A., and Shevchenko, I.: Physical and chemical properties of deposited airborne particulates over the Arabian Red Sea coastal plain, Atmospheric Chemistry and Physics, 17, 11467-11490, https://doi.org/10.5194/acp-17-11467-2017, 2017.

Marsden, N. A., Flynn, M. J., Allan, J. D., and Coe, H.: Online differentiation of mineral phase in aerosol particles
135 by ion formation mechanism using a LAAP-TOF single-particle mass spectrometer, Atmospheric Measurement Techniques, 11, 1, 195-213, 10.5194/amt-11-195-2018, 2018.

Marsden, N. A., Ullrich, R., Möhler, O., Hammer, S. E., Kandler, K., Cui, Z., Williams, P. I., Flynn, M. J., Liu, D., Allan, J. D., and Coe, H.: Mineralogy and mixing state of north African mineral dust by online single-particle mass
140 spectrometry, Atmospheric Chemistry and Physics, 19, 2259-2281, https://doi.org/10.5194/acp-19-2259-2019, 2019.

Ott, D. K., and Peters, T. M.: A Shelter to Protect a Passive Sampler for Coarse Particulate Matter, $PM_{10-2.5}$, Aerosol Science and Technology, 42, 4, 299-309, 10.1080/02786820802054236, 2008.

Prakash, P. J., Stenchikov, G., Tao, W., Yapici, T., Warsama, B., and Engelbrecht, J. P.: Arabian Red Sea coastal
145 soils as potential mineral dust sources, Atmospheric Chemistry and Physics, 16, 18, 11991-12004, doi:10.5194/acp-16-11991-2016, 2016.

---

## Author Comment (AC1)

The authors gratefully acknowledge the reviewers for their valuable comments. Below we provide detailed responses to address each reviewer's comment separately. Our responses to reviewer comments are in blue normal text and changes in manuscript in green normal text.

In the revised paper, an additional table (Table 1) causes latexdiff to not give any marked-up changes. Therefore, the tracked change file is made without including Table 1.

List of all relevant changes made in the manuscript

- **A table detailing size-resolved particle composition and particle type by both number and mass**

- **New Supplement with table including classification scheme, size resolved elemental mass fraction, and average elemental composition (%) as a function of particle size for different particle groups**

- **The sections Abstract, Methodology and Conclusion have been updated according to the comments of the reviewers**
* * *
**Anonymous Referee #1**

**General comments**

The manuscript describes the sampling and analysis of mineral dust samples collected in the Moroccan Sahara in September 2019. For inter-comparison the dust particles were collected by three sampling methods, including flat-plate sampler (FPS), free-wing impactor (FWI), and a multi-stage deposition impactor (MOUDI). Real time particle size distributions were also measured by optical particle counter (OPC). The samples were analyzed by automated scanning electron microscopy (SEM), generating analyses on particle morphology and chemical composition, of more than 300,000 individual dust particles. Normalized elemental results were assembled into eight particle classes by a set of rules, each class approximating a likely dust mineral. These were labelled as hematite-like, quartz-like, feldspar-like, calcite-like, gypsum-like, halite-like ammonium sulphate-like, kaolinite-like, etc. The results were classified by particle diameter into eight size bins from, $< 1$ to $128$ $\mu$m, each size bin containing the particle counts in that size bin. Inter-comparative plots of number abundances of the mineral-like and other chemical parameters abundances are provided. Discussion of the iron comparative number distribution plots for the FPS, FWI, two stages of the MOUDI, as well as for the OPC are given, with discussion of the mixing state thereof. Aspect ratios are compared with those from other studies.

The title clearly reflects the content of the paper and the abstract provides a concise summary. It adequately addresses the in situ sampling and SEM analysis of mineral dusts, of importance to a better understanding of airborne minerals and their potential impact on Global climate. Although automated SEM analysis has been known for more than two decades (Engelbrecht et al., 2009), the application thereof to dust analysis remains novel, especially regarding the interpretation of data. Research presented in this paper further explores the automated SEM procedures and technology. A Global library of airborne dust mineralogical and morphological data will facilitate the interpretation of satellite signals. To readily compare with other similar studies, the chemical results should be calculated to reflect mass fractions or mass percentages, not number abundances.

**Specific comments**

1. Particle **number** abundances (fractions) per "volume equivalent diameter" are presented in figures and text in the manuscript. In a few instances (lines 16, 447, 532, 574) the authors do refer to volume fractions. As shown in the below figures (Engelbrecht et al., 2017), there is a substantial bias towards the number of smaller particles. To better compare with other bulk analytical techniques, individual particle **volume** or **mass** (Ott and Peters, 2008a; Prakash et al., 2016; Engelbrecht et al., 2009; Engelbrecht et al., 2017; Marsden et al., 2018; Marsden et al., 2019; Ott and Peters, 2008b) can be assessed from SEM based particle area measurements (Engelbrecht et al., 2009; Prakash et al., 2016; Ott and Peters, 2008b). Similarly, for intercomparison of chemical parameters between particle size bins, it is preferable to express abundances as equivalent **volumes**, and if particle densities are known, particle mass.

It is recommended that the number concentrations be converted to at least volume concentrations throughout this manuscript, and that the chemical parameters be assessed in terms of volume fractions or volume percentages. The figures illustrating the chemical parameters will need to be re-compiled to reflect volume (or mass) concentrations and not number abundances. In this way the results can be readily compared to other SEM based studies, as well as to chemical results generated by other analytical techniques such as SPMS, ICP-MS, XRF, etc.

In the revised manuscript the number fractions are converted to mass fractions by assuming spherical particles and assigning an according bulk density estimated from composition.

2. Results from this study should be discussed in conjunction with individual particle dust studies from other desert regions, including North Africa and the Middle East (Engelbrecht et al., 2009; Engelbrecht et al., 2017; Prakash et al., 2016).

The text has been edited to include information about individual particle dust studies from other desert regions.

**Some Technical Corrections to consider**

Line 43.properties and mineral composition

Changed as requested.

Line 47. deposit faster due to gravity, . . . .

Changed as requested.

Line 53. ..dust is more optically absorbing when. . .

Changed as requested.

Line 57. ..not only on the source composition but also on the feldspar. . .

Changed as requested.

Line 80. . . . are the interdependencies among particle size, composition,. . .

Changed as requested.

Line 82. . . . mixing state upon both particle composition and size. . .

Changed as requested.

Line 83. . . . mineralogical composition of windblown dust are relatively new . . .

Changed as requested.

Line 89. Unclear, rephrase sentence. "One reason for such measurement scarcity is that source areas, particularly the host prolific ones, are typically located in remote areas and subject to harsh conditions."

Rephrased to,'However, despite its importance, there exist very few ground based in situ measurement studies to characterize the particle composition and microphysical properties. This is mainly due to the difficulty in dust sampling at source, as source areas are often remote and have harsh environment.'

Line 91. Dust storms result in high particulate concentrations and filter overload, providing a challenge to analyze by automated SEM.

Changed as requested.

Line 96. Three different instruments were used to sample mineral dusts, for the measurement of chemical and physical properties of individual dust particles by SEM coupled with an energy-dispersive X-ray analyzer (EDX).

Changed as requested.

Line 99. ... individual dust minerals to ultimately help understand...

Changed as requested.

Line 101. Such information is required to better constrain climate models that consider mineralogical variations in their representation of the dust cycle (Perlwitz et al., 2015a; Scanza et al., 2015; Li et al., 2021). This is timely, given the prospect of global soil mineralogy retrievals using high-quality satellite hyperspectral measurements (Green et al., 2020).

Changed as requested.

Line 117. Under favorable weather conditions, dust is frequently emitted from this source area. (GonzálezFlórez et al., in prep.).

Changed as requested.

Line 129. ...FWI were collected twice daily with a typical....

Changed as requested.

Line 138. ...particle size cutoff is determined by the impaction ...

Changed as requested.

Line 145. The main controls for particle ...

Changed as requested.

Line 153. ...and fourth stages were selected for detailed analysis.

Changed as requested.

Line 154. ..when particles impact on the collection substrate....

Changed as requested.

Line 156. ....depending on the mineralogical properties of particles....

Changed as requested.

Line 162. Previous studies (Kandler et al., 2008........

Changed as requested.

Line 162. Add references (Engelbrecht et al., 2009; Engelbrecht et al., 2017; Prakash et al., 2016)

Thank you for these references. They have been added.

Line 166. ....without any pretreatment. Particles were automatically detected and analyzed .......

Changed as requested.

Line 167. ...(BSE) imagery was used for particle detection, as dust particles ....

Changed as requested.

Line 170. .. number of input counts by the EDX detector.

Changed as requested.

Line 174. Accounting for matrix dependent efficiencies of the....

Changed as requested.

Line 176. The SEM-EDX results are normalized to 100 %, including those of the elements C, N, O, Na, Mg, Al, Si, P, S, Cl, K, Ca, V, Cr, Mn, Fe, Zn, and Pb. ....... Commnent: What is the minimum particle size measured by automated SEM-EDX? Changed as requested. The minimum size measured in projected area diameter is 0.5 µm for FPS and MOUDI and 2.5 µm for FWI. This is mentioned in line 176 in the revised version.

Line 228. SEM-EDX measures the elemental composition

Changed as requested.

Line 229. ...can therefore be readily identified (e.g. gypsum, quartz, calcite).

Changed as requested.

Line 238. X is defined as the ratio of the mass concentration of the element considered to the sum of the mass concentrations of all the measured elements.

Changed as requested.

Line 243. ...method was not applied to quantify...

Changed as requested.

Line 246. Their most prevalent chemical components, ...

Changed as requested.

Line 247. The mineral labels were assigned from the most prevalent measured elemental concentrations. There was no actual phase identification ......

Changed as requested.

Line 252. Aerosol PSDs of suspended mineral dusts were also obtained by Optical Particle Counter (OPC).......

Changed as requested.

Line 264. .....only error considered is the Poisson counting error. Explain or delete sentence

Sentence removed, and is only mentioned in the figure caption.

Line 268 Based on set classification rules, particles. . . .

Changed as requested.

Line 275 . . . particle types were found mainly in particles with. . .

Changed as requested.

Line 284. . . . .Fe content of quartz-like particles is generally low and variable, suggesting that Fe oxides are not an integral part of the quartz-like particles.

Changed as requested.

Line 289. Ca-feldspar-like particles are quite rare in the samples. Explain why, plagioclases are seldom pure albite, often oligoclase or andesine, depending on their provenance. This anomaly perhaps to do with the SEM analytical procedure?

Larger plagioclase grains usually tend to be aggregates with some amount of coating present. Even though the structural properties cannot be studied by SEM analytical procedure, we have presented an index approach where the feldspar composition is calculated to that of an ideal composition.

Line 301. Figure 2 y-axis. Are relative abundances by number or volume (or mass)?

The relative abundance is by number. For consistency, the relative abundance is updated to include average weighted by mass.

Line 326. Ammonium sulfate is the prevalent sulfate species in atmospheric aerosols and generally of anthropogenic origin.

Changed as requested.

Line 373. On average across all particle sizes, by number, 26% of particles are illite-like, . . . . . .

The discussion is updated and includes relative abundance by mass. See reply to specific comment 2.

Line 416. . . . . along the air stream onto the lower stages.

Changed as requested.

Line 422. While the dependence of particle size distribution on sample mineralogy is quite strong, the temporal . . .

Changed as requested.

Line 431. There are fertilizers that do not contain K, such as di-ammonium phosphate (DAP), mono-ammonium phosphate (MAP). Hydroxy apatite is not commonly found in the Morocco phosphate deposits, the mined francolite generally contains F.

We changed the statement to, "Apatite-like particles were almost non-existent in our samples except for some specific days (20 and 23 September). They were identified by the presence of Ca and P. Furthermore, manual inspection of the particles showed it to be not beam-sensitive with no Cl and F signals detected. Even though hydroxyapatite $[Ca_5OH(PO_4)_3]$ is not commonly found in the Morocco phosphate deposits, the absence of Cl and F suggests it to be OH-apatite originating from a distinct geogenic source."

Line 439. Figure 8. Phase compositions of size bins for samples collected under specific meteorological conditions at specific times and on certain days.

Caption changed as requested.

Line 439. The total number of analyzed particles is given for…..

Typo corrected.

Line 441. feldspar not a typical Fe bearing mineral, only K-feldspar in very small amounts.

Indeed. We rephrased it to, "Clay minerals and Fe-oxi/hydroxides such as haematite, goethite and magnetite..."

Line 474, Figure 9. Relative number abundances?

It has now been updated to reflect relative mass fraction. Consequently, the caption and y-axis label has been revised to make this clear.

Engelbrecht, J. P., McDonald, E. V., Gillies, J. A., Jayanty, R. K. M., Casuccio, G., and Gertler, A. W.: Characterizing mineral dusts and other aerosols from the Middle East – Part 1: Ambient sampling, Inhalation Toxicology, 21, 4, 297-326, doi: 10.1080/08958370802464273, 2009.

Engelbrecht, J. P., Stenchikov, G., Prakash, P. J., Lersch, T., Anisimov, A., and Shevchenko, I.: Physical and chemical properties of deposited airborne particulates over the Arabian Red Sea coastal plain, Atmospheric Chemistry and Physics, 17, 11467-11490, https://doi.org/10.5194/acp-17-11467-2017, 2017.

Marsden, N. A., Flynn, M. J., Allan, J. D., and Coe, H.: Online differentiation of mineral phase in aerosol particles by ion formation mechanism using a LAAP-TOF single-particle mass spectrometer, Atmospheric Measurement Techniques, 11, 1, 195-213, 10.5194/amt-11-195-2018, 2018.

Marsden, N. A., Ullrich, R., Möhler, O., Hammer, S. E., Kandler, K., Cui, Z., Williams, P. I., Flynn, M. J., Liu, D., Allan, J. D., and Coe, H.: Mineralogy and mixing state of north African mineral dust by online single-particle mass spectrometry, Atmospheric Chemistry and Physics, 19, 2259-2281, https://doi.org/10.5194/acp-19-2259-2019, 2019.

Ott, D. K., and Peters, T. M.: A Shelter to Protect a Passive Sampler for Coarse Particulate Matter, $PM_{10-2.5}$, Aerosol Science and Technology, 42, 4, 299-309, 10.1080/02786820802054236, 2008.

Prakash, P. J., Stenchikov, G., Tao, W., Yapici, T., Warsama, B., and Engelbrecht, J. P.: Arabian Red Sea coastal soils as potential mineral dust sources, Atmospheric Chemistry and Physics, 16, 18, 11991-12004, doi:10.5194/acp-16-11991-2016, 2016.

**Anonymous Referee #2**

**Overview**

This work describes SEM-EDX and particle size analysis of freshly collected mineral dust from the Moroccan desert. The authors combine multiple collection methods and describe how mineralogical composition changes over the particle size distribution. Measurements of mineralogical composition in mineral dust aerosol are still sparse, so the dataset presented by the authors is valuable and relevant to ACP. The work suffers from issues in presentation and does not always transition from data presentation to the expected interpretation. I recommend that this work be accepted only after the authors revise the work.

**Overall Comments**

1. The authors often spend a significant amount of time in the results citing previous work but not directly connecting the discussion to the current analysis in a clear way. This work will be significantly more impactful if the authors work on conciseness.

We have now moved the particle classification part in the supplement and updated the discussion by including comparison with other source region.

2. The mineral identification is very roughly explained, and no validation is presented. At a minimum, citations need to be included showing that the development of user ID'd mineralogical classes is an appropriate method for these data. If the authors were only presenting a few major classes, what's presented may be acceptable, but 22 mineral classes require some validation. If the authors could present some analysis of known standards, it would greatly strengthen the defensibility of this dataset.

Two things should be distinguished here: the validity of the measurements and the validity of the interpretation. We assume that the reviewer is not questioning the validity of the measurements itself, as they are an established technique (Levin et al. 1996; Reid et al. 2003; Kandler et al. 2007; Coz et al. 2009; Engelbrecht et al., 2009) and many others. These data are published along with the current manuscript (data will be made available in Zenodo), so anyone can later use the particle information for a different interpretation.

Regarding the validity of the interpretation in mineralogy-like classes, there are some classes, which can be clearly distinguished from others (eg., sea-salt, quartz), some which are indistinguishable by our technique in general (eg. Gypsum/anhydrite, rutile/anatase), and some which may contain ambiguous compositions and therefore are prone to a potential classification error (eg., micas, smectites).

Based on this situation, we try our best to refine our chemical classification to the range known for the composition of the mineral (or better: mineral group) classes, in which we sort them. However, we are aware that this is not a true mineralogy, for which reason we label the classes as 'mineral-like'.

Unfortunately, there is no 'known standard' as requested, suitable for the electron microscopic approach. There are XRD-characterized materials, but they are unsuitable for a single particle analysis, as they are bulk samples and suffer size statistics problems on single particle level. And there are pure minerals (e.g as TEM standards), but they commonly contain one perfectly crystalline material. For these samples, the approach works well, but it is not generalizable to a natural sample. So the only option by now is checking for plausibility. We have analyzed a

sediment derived from physical weathering from granite and compared for XRD (unpublished). Here, our approach works well, but a desert soil is more complex with respect to ambiguous clay minerals. We have developed our approach from a simple scheme (Kandler et al. 2007; Kandler et al. 2009) to more complex ones (Kandler et al. 2011; Kandler et al. 2018). In comparing different desert aerosols (Kandler et al. 2020), we found plausible differences related to geology and weathering conditions, but in particular between different techniques (Marsden et al. 2018) there will remain biases. We have updated section 2.6 in the revised paper addressing part of these discussion.

3. The change in mineral composition/size distribution of particles from soil to suspended mineral dust is an important point. It would improve the paper to include a direct comparison to the soil data here (not just reference to a forthcoming paper).

We absolutely agree with the reviewer, that this work is highly important. In fact this is a key aspect tackled in the the FRAGMENT project, the framework of this study. Understanding the relationship of the emitted size-resolved composition and that of the parent soil is key to include mineralogical dependencies in climate models. However this challenging aspect is beyond the scope of the present work. It requires a considerable amount of laboratory work, as for different soils, different size classes have to be generated. They need to be measured and validated by more than one technique to assess reliability of each technique. And then these samples need to be analyzed by SEM-EDX, which requires another large amount of resources. We have made first steps and analyses in this aspect, but results are not yet ready and will be presented in forthcoming publications. We have highlighted this point in the conclusion.

4. Aggregate data presentation. Throughout the work, the authors present compositional size distributions that seem to have aggregated multiple collection methods with different collection efficiencies, target size ranges, and sampling intervals (for some the sampling intervals are very short.) The authors should more explicitly describe this aggregation process so it can be accessed for validity.

The primary reason for aggregating the data is because the collection efficiency by size is less relevant to the fractional contribution of each mineral type per size and therefore integrating all the techniques together improves our statistics per each size (higher number of particles analysed).

Regarding the sampling intervals, we set up non-parallel sampling time because FPS operates passively while FWI and MOUDI operates actively. The active samplers have a much higher collection velocity compared to the passive one. Therefore, we cannot set up the same time interval for both type of samplers, as this would result in either overloading of the active or underloading of the passive sampler for single particle analysis. Most of the sampling from the active samplers were done simultaneously with similar aerosol conditions. Furthermore, as the target size for the two active samplers is different($< 2.5$ µm for MOUDI and $> 3$ µm for FWI), therefore FWI result are suitable for coarse dust and MOUDI and FPS towards fine dust. We have included these points in Section 3.3 and 3.3.1 of the revised paper.

**Specific Comments**

Supplement: For the particle ID code (the key component of this text), the text should be included, not just a screenshot. The authors should also list the software used here.

The new supplement consists now of the classification table. The morphological and chemical measurements are acquired automatically by AZtecFeature software for particle classification schemes. This information is provided in section 2.3 of the updated paper.

39: replace "catalyzer" with catalyst

Changed as requested.

84: On a global scale, the mineralogical maps presented by Journet et al. 2014 are relatively high resolution and based on a large number of mineralogical measurements, albeit with notable gaps. The authors should instead emphasize the gap in knowledge bridging soil mineralogical and size distributions to those found in dust aerosol, as this is a major challenge.

The most complete dataset available (Jounet et al. 2014) characterizes the clay (0-2 $\mu$m) and silt (2-63 $\mu$m) size ranges of FAO soil types (units) in terms of abundance of 12 key minerals important for dust-climate interactions, i.e., quartz, feldspar, illite, smectite, kaolinite, chlorite, vermiculite, mica, calcite, gypsum, hematite and goethite. While this dataset represents a substantial improvement compared with a previous version (Claquin et al ., 1999), it is based upon barely 700 soil descriptions that sample only 55% of the FAO soil units. Moreover, the mineralogical samples do not cover many regions that are prolific dust sources, and the distribution of records within sampled regions is very inhomogeneous. Given the limited direct measurements of soil mineral content, a global atlas is created by relating mineralogy to the more widely characterized soil type. Therefore there is an extrapolation of a limited amount of observations. Even if soil type is available at relatively high resolution, the associated mineralogy is not. In the near future observations obtained with spectroscopy from space will provide additional observations at high resolution. Our text in the manuscript reflects both the current uncertainties in the global soil atlases of mineralogy and on the limited understanding of the size-resolved mineralogy at emission that is partly tackled in this contribution.

'Models that include spatiotemporal variations in mineralogical composition are relatively new (Perlwitz et al., 2015a, b; Scanza et al., 2015) and currently use rather crude soil mineralogy maps (Claquin et al., 1999; Nickovic et al., 2012; Journet et al., 2014) as a lower boundary condition. Soil mineralogy maps are based on massive extrapolation from a limited amount of soil mineralogical analyses, ancillary information on soil texture and color, and several additional assumptions. This limited knowledge together with our incomplete understanding of the chemical and mineralogical size fractionation between soil and emitted dust hinders our ability to extend and constrain theories of dust emission (Perlwitz et al., 2015a; Pérez García-Pando et al., 2016; Li et al., 2021).'

2.7: The authors should note that Optical particle counters usually rely on assumed refractive indexes and particle shapes in order to determine distributions. Thus, in a dust-dominated system with irregular particle shapes, the OPC is likely to be highly uncertain. This should be, at minimum, discussed and perhaps adjusted for in the discussion of the data.

Indeed. We have now updated the figure to reflect this. Furthermore, in our companion paper González-Flórez et al., 2022 we have discussed it in more detail. As the OPC size distribution in this paper is used to compare the shape of the distribution and not the absolute concentration, we have not discussed it in detail.

2.8 Is the confidence interval simply the confidence of the fit of the binomial distribution to the observed particle size distribution, or are other uncertainties included?

The confidence interval based on a binomial distribution is used as an estimate for the relative abundance. Other uncertainties like measurement errors in the signal quantification are not included.

307: What is the typical percentage or range of Ca-rich particles in the larger size classes? For

other types, percentages or ranges were included.

The Ca-rich group, represents about 10 % for $d_v < 2.5$ $\mu$m and around 5 % for $2.5 < d_v < 10$ $\mu$m. This information is now on section S 2 in the supplement. Furthermore, Table 1 in the revised paper show the size resolved mass and number fraction (%) of particle class.

3.1: It would be useful to the reader to see the percentages of particles in a table (average, range, median, etc) for each compositional class and size type. The size classes referenced in the text are inconsistent between compositional classes making comparisons difficult. This may be suitable for the supplement.

We have now have included a table in section 3.3 showing the size resolved mass and number fraction (%) of particles in each particle class. In addition, a table showing the relative fraction of particles with 95 % confidence interval is provided in the supplement.

373: I would caution against presenting an average of percentages over the different size classes. This average loses information such as particle number concentration and masks compositional differences between size classes.

Agreed. We have now modified this discussion section.

375: Do you believe the ammonia sulfate to be soil/dust sourced? It seems likely that this is secondary aerosol, which would support its presence in the small-size bins.

We interpreted it to be secondary as it was present only in few of our samples and especially during 8-10 September when the wind comes from the east (i.e. M'Hamid) suggesting anthropogenic origin which is further corroborated with measurements of optical properties (Yus-Díez et al., in prep.) and PSD (González-Flórez et al., 2022). This is mentioned in Line 426 of the original manuscript. Looking at the composition of particles, it could have been of marine origin at some point, and anthropogenically processed and fractionated. In the ternary plot below, the color scale shows the relative ion balance [defined as SSRIB=(Na+2*Mg+K+2*Ca-Cl-2*S)/(Na+2*Mg+K+2*Ca+Cl+2*S)] runs basically from 0 to 0.3 i.e., from neutral to missing cations, which could be nitrate.

[Figure]

380: The change in mineral composition/size distribution of particles from soil to suspended mineral dust is an important point. It would improve the paper to include a comparison to the soil data here (not just reference to a forthcoming paper).

See our answer in the Overall comments point 3.

380-393: This presentation reads like a collection of unrelated thoughts. I recommend rewriting to emphasize how this puts your data into context.

We have rewritten this part.

Figure 4: Please explain what "normalized" means in the left axis. What are the data normalized to? How is this different that dn/dlog(d), and why is it used?

Our comparison focuses on the PSD (not the total amount) and therefore for comparison we normalize. The size distributions were normalized by fitting the number (N) size distribution to the power law $dN/d \log D_p = cD_p^{-2}$, and dividing all measurements by the fitted proportionality constant c (Kok, 2011). We updated section 2.5.2. to reflect this.

3.3: It seems an important point to parse whether larger particles are mostly aggregates of internal mixed smaller particles or have uniform composition. There is a good discussion in this section, but I wonder if additional analysis could be performed. Could you compare the elemental ratios Fe allocated to the "complex" categories oversize classes to gain some insight into this? For example, does the Fe:Al ratio go up with size class for these complex categories? This would imply the internal mixing of high Fe particles. A "simple" two-source mixing model maybe could be used to estimate the Fe-oxi/hydroxides vs. structural iron. I'm not confident using the Si:Al ratio as a filter as you do in this section gives a good reference point. As your major iron hydroxides (hematite, Goethite) don't contain Si, this just allows you to compare siliceous, and clay minerals, which I'm not convinced is a useful comparison for Fe.

One of our aims in this section is to look at specifically the iron inclusion integrated in particles. In Fig.5 of the preprint, we show the evolution of the different particle classes over size including Fe-rich particles. As dust is primarily composed of silicates, we use the Si:Al ratio for reasons described in section S 2.9 in the supplement to get an indication of iron distribution on the silicate particles. We also compare the particle types suggested by the reviewer including the Fe-rich particle type. Two of the "complex" categories have feldspar-like composition with 0.25<Al/Si< 0.5 and the complex-clay mineral like has Al/Si > 0.5. As all of the hematite-like particles have an iron index $(|Fe|) > 0.50$ and the relative fraction of them is significantly less than that of aluminosilicates and clay minerals, its contribution in rest of the iron indices is not relevant. This is relatively well captured by our analysis as most of the contribution with high iron inclusion are from clay minerals. And the fraction with Al/Si < 0.1 and $(|Fe|) > 0.50$ is only seen in particles with diameter < 4 $\mu$m which could be complex quartz-like particles with some free Fe (oxy-hydr)oxides. In fact, the major Fe-rich particle type i.e., hematite does contain Si on average as seen in Fig. 2. We have plotted the elemental ratio as requested by the reviewer and further included Fe/Si and Al/Si as it makes for the major part of the particles. We do this but don't think it adds new insights to the discussion of iron than already presented.

[Figure]

Mean elemental ratio for selected particle types.
The error bars are standard error and have been propagated.

Coz, E., F. J. Gómez-Moreno, et al. (2009): Individual particle characteristics of North African dust under different long-range transport scenarios. Atmos. Environ. 43, 1850-1863.

Kandler, K., N. Benker, et al. (2007): Chemical composition and complex refractive index of Saharan Mineral Dust at Izaña, Tenerife (Spain) derived by electron microscopy. Atmos. Environ. 41(37), 8058-8074. doi: 10.1016/j.atmosenv.2007.06.047

Kandler, K., K. Lieke, et al. (2011): Electron microscopy of particles collected at Praia, Cape Verde, during the Saharan Mineral dust experiment: particle chemistry, shape, mixing state and complex refractive index. Tellus 63B, 475-496. doi: 10.1111/j.1600-0889.2011.00550.x

Kandler, K., K. Schneiders, et al. (2018): Composition and mixing state of atmospheric aerosols determined by electron microscopy: method development and application to aged Saharan dust deposition in the Caribbean boundary layer. Atmos. Chem. Phys. 18, 13429-13455. doi: 10.5194/acp-18-13429-2018

Kandler, K., K. Schneiders, et al. (2020): Differences and Similarities of Central Asian, African, and Arctic Dust Composition from a Single Particle Perspective. Atmosphere 11(3), 269. doi: 10.3390/atmos11030269

Kandler, K., L. Schütz, et al. (2009): Size distribution, mass concentration, chemical and mineralogical composition, and derived optical parameters of the boundary layer aerosol at Tinfou, Morocco, during SAMUM 2006. Tellus 61B, 32-50. doi: 10.1111/j.1600-0889.2008.00385.x

Levin, Z., E. Ganor, et al. (1996): The Effects of Desert Particles Coated with Sulfate on Rain

Formation in the Eastern Mediterranean. J. Appl. Meteorol. 35, 1511-1523. doi: 10.1175/1520-0450(1996)035<1511:TEODPC>2.0.CO;2

Marsden, N. A., M. J. Flynn, et al. (2018): Online differentiation of mineral phase in aerosol particles by ion formation mechanism using a LAAP-TOF single-particle mass spectrometer. Atmos. Meas. Tech. 11(1), 195-213. doi: 10.5194/amt-11-195-2018

Reid, E. A., J. S. Reid, et al. (2003): Characterization of African dust transported to Puerto Rico by individual particle and size segregated bulk analysis. J. Geophys. Res. 108(D19), 8591. doi: 10.1029/2002JD002935

González-Flórez, C., Klose, M., et al. (2022): Insights into the size-resolved dust emission from field measurements in the Moroccan Sahara, Atmospheric Chemistry and Physics Discussions, 2022, 1–65, https://doi.org/10.5194/acp- 2022-758, 2022

Kok, J. F. (2011). A scaling theory for the size distribution of emitted dust aerosols suggests climate models underestimate the size of the global dust cycle. Proceedings of the National Academy of Sciences, 108(3), 1016-1021.

Waza, A., Schneiders, K., et al. (2019a): Field comparison of dry deposition samplers for collection of atmospheric mineral dust: results from single-particle characterization. Atmospheric Measurement Techniques, 12(12), 6647-6665.

Waza, Andebo; Schneiders, Kilian; Kandler, Konrad (2019b): Daily dust deposition fluxes at Izana, Tenerife collected by different techniques: particle size and composition from single particle electron microscopy. PANGAEA, https://doi.org/10.1594/PANGAEA.901413

Kandler, Konrad; Lieke, Kirsten; Schneiders, Kilian; Heuser, Johannes (2019a): Microphysics and chemical composition of particulate dry deposition measured at Fogo, Cape Verde. PANGAEA, https://doi.org/10.1594/PANGAEA.903322

Kandler, Konrad; Deutscher, Carmen; Lieke, Kirsten; Schütz, Lothar (2019b): Microphysics and chemical composition of particulate dry deposition measured in Praia, Santiago, Cape Verde. PANGAEA, https://doi.org/10.1594/PANGAEA.903325

---

## Author Response (AR2)

Dear Prof. Bahreini,

Thank you for the assessment of our manuscript for publication in ACP. We have modified and revised the manuscript to address the editorial changes. The line number in our response refers to the track-changed revised version.

On behalf of my co-authors

Agnesh Panta
* * *
L95- Change to "However, despite their importance, very few ground-based, in-situ measurement studies to characterize the particle composition and microphysical properties exist. ....Also, frequent dust storms result in high particulate concentrations and filter overload, which post a challenge for these samples to be analyzed by automated SEM."

Changed as requested. L93-96

L247: Change to "....related to geology and weather conditions were observed; however, various biases also exist between different techniques (Marsden et al., 2019)."

Changed as requested. L241-242

L293: Table number is missing

We suppose you mean L393. As it was also mentioned in our previous response, inclusion of Table 1 causes latexdiff to not give any marked-up changes. Therefore, in the tracked change version Table 1 was not included. It appears in the revised version. L305

L396-397: either delete "as" or "and therefore"

"as" is deleted. L308

L407: Change to "... and their contribution decreases with particle size.."

Changed as requested. L318

L409: Change to "... iron is detected across all the size classes...."

Changed as requested. L320

L442: Capitalize Middle East

Changed as requested. L341

L682: change "tacked" to "investigated"

Changed as requested. L554